DOI: 10.1038/s41467-017-01326-5　　**OPEN**

# Evolution of AF6-RAS association and its implications in mixed-lineage leukemia

Matthew J. Smith [1,2], Elizabeth Ottoni[1], Noboru Ishiyama[3], Marilyn Goudreault[1], André Haman[1], Claus Meyer[4], Monika Tucholska[5], Genevieve Gasmi-Seabrook[3], Serena Menezes[3], Rob C. Laister[3], Mark D. Minden[3,6], Rolf Marschalek [4], Anne-Claude Gingras[5,7], Trang Hoang[1,8] & Mitsuhiko Ikura[3,6]

Elucidation of activation mechanisms governing protein fusions is essential for therapeutic development. *MLL* undergoes rearrangement with numerous partners, including a recurrent translocation fusing the epigenetic regulator to a cytoplasmic RAS effector, AF6/afadin. We show here that AF6 employs a non-canonical, evolutionarily conserved α-helix to bind RAS, unique to AF6 and the classical RASSF effectors. Further, all patients with MLL-AF6 translocations express fusion proteins missing only this helix from AF6, resulting in exposure of hydrophobic residues that induce dimerization. We provide evidence that oligomerization is the dominant mechanism driving oncogenesis from rare *MLL* translocation partners and employ our mechanistic understanding of MLL-AF6 to examine how dimers induce leukemia. Proteomic data resolve association of dimerized MLL with gene expression modulators, and inhibiting dimerization disrupts formation of these complexes while completely abrogating leukemogenesis in mice. Oncogenic gene translocations are thus selected under pressure from protein structure/function, underscoring the complex nature of chromosomal rearrangements.

[1] Institute for Research in Immunology and Cancer, Université de Montréal, Montréal, QC H3T 1J4, Canada. [2] Department of Pathology and Cell Biology, Faculty of Medicine, Université de Montréal, Montréal, QC H3T 1J4, Canada. [3] Campbell Family Cancer Research Institute, Princess Margaret Cancer Centre, Toronto, ON M5G 2C1, Canada. [4] Institute of Pharmaceutical Biology, Goethe-University of Frankfurt, Frankfurt, 60323, Germany. [5] Lunenfeld-Tanenbaum Research Institute, Mount Sinai Hospital, Toronto, ON M5G 1X5, Canada. [6] Department of Medical Biophysics, University of Toronto, Toronto, ON M5G 1L7, Canada. [7] Department of Molecular Genetics, University of Toronto, Toronto, ON M5S 1A8, Canada. [8] Department of Pharmacology, Faculty of Medicine, Université de Montréal, Montréal, QC H3T 1J4, Canada. Correspondence and requests for materials should be addressed to M.J.S. (email: matthew.james.smith@umontreal.ca) or to M.I. (email: mikura@uhnres.utoronto.ca)

Chromosomal translocations arising from genomic instability result in gene fusions that drive an estimated 20% of human cancers, predominantly hematological malignancies[1]. Rearrangements of the mixed-lineage leukemia gene (*MLL*; also *MLL1* or *KMT2A*) result in acute myeloid and lymphoid leukemia's in both adults and children, cancers associated with poor prognosis and which urgently require new therapeutic strategies[2]. *MLL* encodes a large histone methyltransferase involved in regulating expression of *HOX* genes[3]. Translocations of *MLL* in hematopoietic cells result in fusions comprising the N-terminal domain of MLL and a C-terminal region encoded by myriad partner genes[4]. Thus, MLL fusions lack the native SET domain and, consequently, H3K4 methyltransferase activity. The divergent properties of fusion partners and the mechanism by which they activate MLL have been an area of intense study for over two decades, but how such a diverse set of partners are able to activate MLL epigenetic activity remains an open question.

The t(6;11)(q27;q23) chromosomal translocation fuses *MLL* to the *AF6* gene[5], creating a fusion implicated in adult, pediatric and infant AML, pro-B ALL and adult T-ALL[6–10]. Expressed is a 334 kDa nuclear protein comprising the N-terminus of MLL and most of AF6 (Fig. 1a). A putative RAS GTPase effector, AF6 is unique amongst highly recurrent MLL partners in that it normally functions outside the nucleus at sites of cell–cell contact[11–14]. It has been demonstrated that AF6-mediated MLL activation is dependent on its first RAS association (RA) domain[15], and differentially tagged MLL-AF6 proteins can be co-precipitated from cell lysates. This suggests transcriptional activation may result via higher order oligomerization, categorizing the recurrent MLL-AF6 among a subset of fusions believed to be mediated via this mechanism. Indeed, the first evidence that

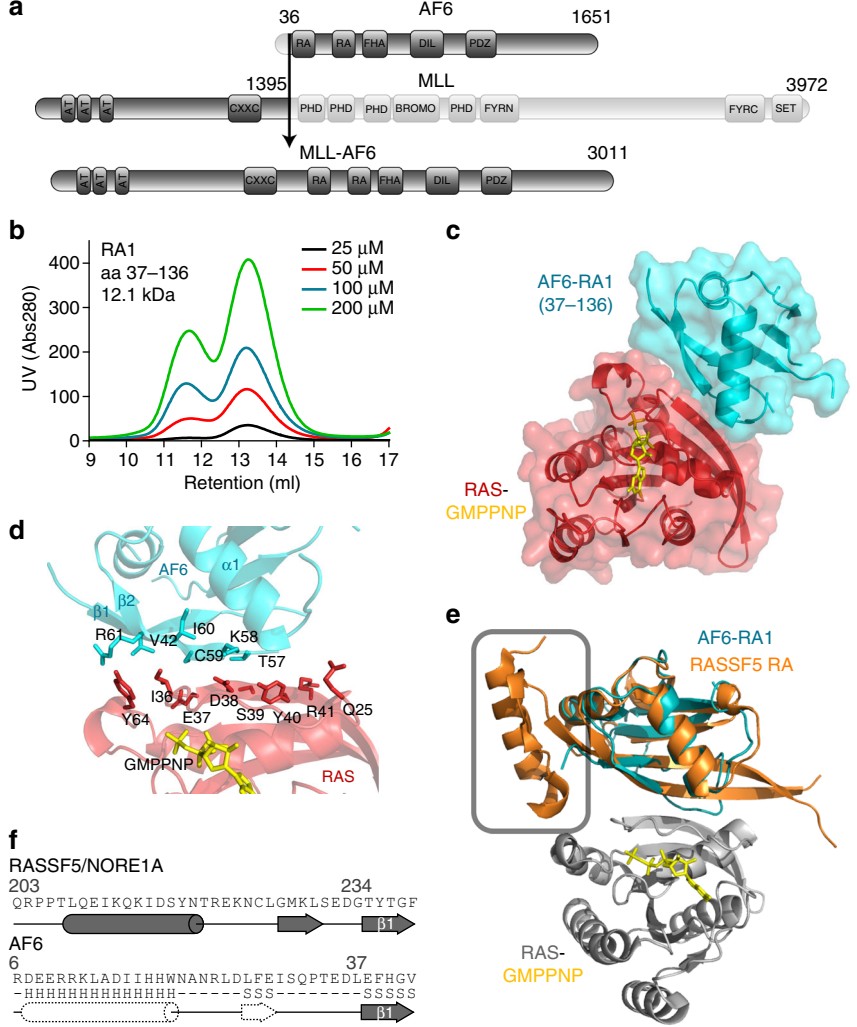

**Fig. 1** Structure of the AF6 RA1 domain complex with activated RAS. **a** The RAS effector AF6 is commonly fused to the epigenetic regulator MLL following a translocation that initiates leukemogenesis. The detected fusion protein connects the N-terminal 1395 residues of MLL to residues 36–1651 of AF6. **b** Size exclusion chromatography of purified AF6 RA1 domain (residues 37–136) presents two peaks, corresponding to monomer and dimer fractions, at several concentrations. **c** Ribbons diagram and surface overlay for crystal structure of the AF6 RA1 domain (blue) complexed with RAS (red) bound to GMPPNP (yellow). Statistics in Supplementary Fig. 2c. **d** Ribbons diagram shows the binding interface between AF6 RA1 domain (blue) and RAS-GMPPNP (red–yellow). Contact residues are labeled and displayed as sticks. **e** Structural alignment of AF6 RA1 domain (blue) complexed with RAS-GMPPNP (gray) and RASSF5 RA domain (orange) bound to RAS-GMPPNP (PDB 3DDC). RASSF5 is the only known RAS effector with an αN helix that contacts the switch II region of RAS. **f** RASSF5 contacts the RAS switch II region via an α-helix located N-terminal to the β1-strand of the RA domain (top, from PDB 3DDC). Predictions of secondary structure upstream of β1 from the AF6 RA1 domain suggest the presence of an analogous helix (bottom, dashed). H = α-helix, S = β-strand (JPRED)

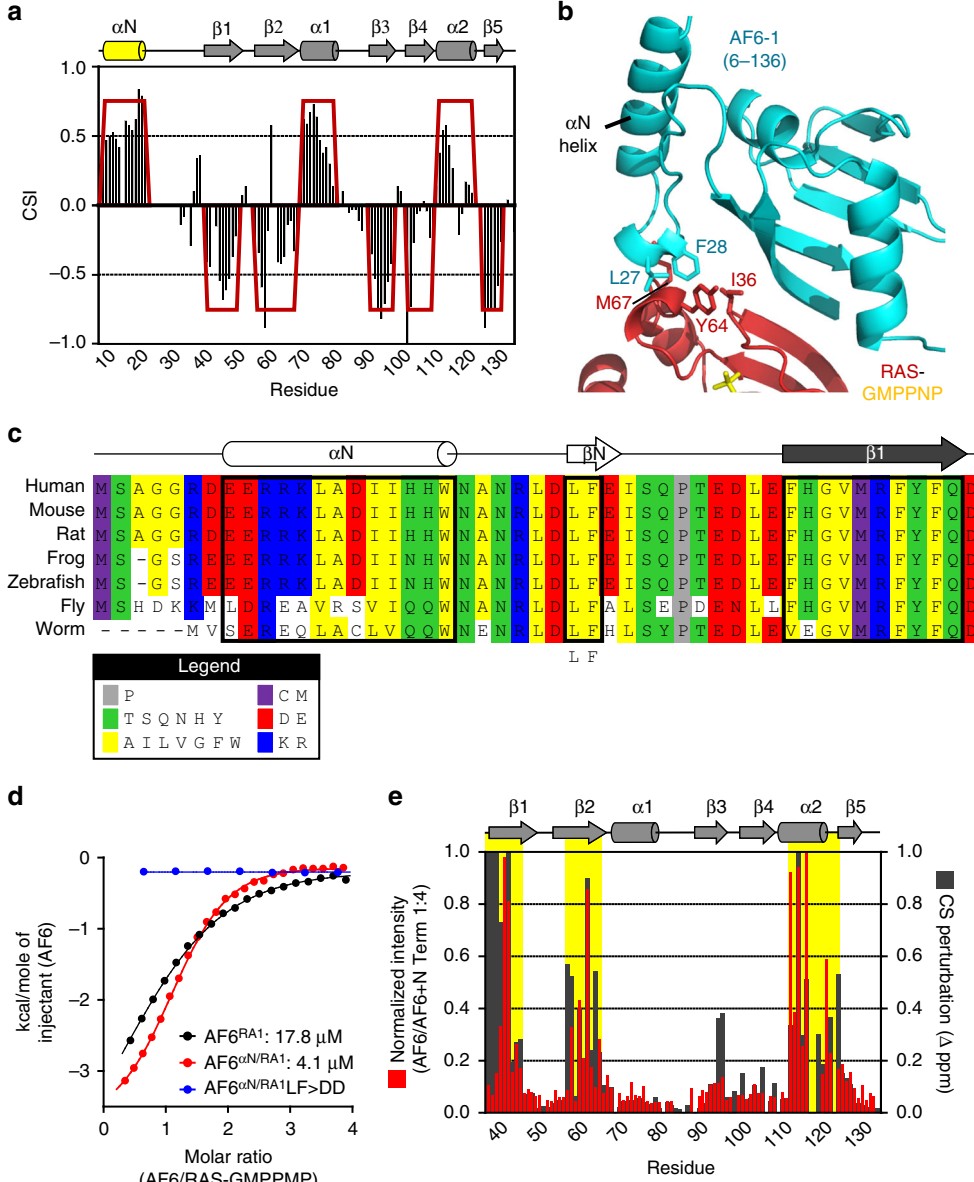

**Fig. 2** The αN helix is present in AF6 and augments its binding to RAS. **a** CSI versus residue number for the AF6 RA1 domain extended 31 residues at the N terminus (6–137). Four positive CSI values indicate α-helix; four negative values indicate β-strand. Resulting secondary structure arrangement depicted at top reveals the N-terminal α-helix (yellow). **b** Ribbons diagram depicting our homology model of the AF6 RA1 domain and αN helix (blue) complexed with RAS-GMPPNP (red-yellow) based on structural alignment with RASSF5-RAS (PDB 3DDC). Leu27 and Phe28, bottom of αN helix, contact switch II residues Met67, Tyr64, and Ile36 of RAS (marked, sticks). **c** Amino acid alignment of residues upstream of the core RA1 domain in seven evolutionarily conserved AF6 orthologues. A Leu-Phe motif in the loop between αN and β1 is completely conserved (boxed). **d** ITC analyses of the interactions between AF6 and RAS-GMPPNP. Core RA1 domain binds RAS with a $K_d$ of 17.8 μM (black), an order of magnitude weaker than most RAS effector interactions. The αN-extended domain exhibits a 4.5-fold increase in affinity (red; $K_d$ of 4.1 μM), while a L27D/F28D double mutant showed no heats of association (blue). **e** Experimental validation of the model by two NMR-based approaches. Left axis, normalized chemical shift perturbations induced in $^1H/^{15}N$-HSQC spectra of the AF6 RA1 domain by a purified AF6 fragment encompassing the αN helix (6–36). Right axis, combined chemical shift differences (Δp.p.m.) of backbone $^1H/^{15}N$ resonances from the core RA1 domain (37–136) vs extended RA1 domain that incorporates the αN helix (6–136). The αN helix in either experiment affects the same three regions (yellow), indicated in domain diagram at top

dimerization can activate MLL came serendipitously, via observation that *Mll-lacZ* caused leukemia in mice[16], and MLL dimerization has been demonstrated as sufficient to immortalize hematopoietic cells[17,18]. Rare MLL fusion partners GAS7[17,19], AF1p/EPS15[17], SEPT6[20], EEN/SH3GL1[21] and GPHN[22] have predicted dimerization motifs indispensable for evoking MLL epigenetic activity. There are no available structural or biophysical data for these regions from any oligomeric fusion partner, but all five of these are predicted to form coiled coils. Similarly,

there is no structure of the AF6 RA domain, a possible mechanism of oligomerization has not been described, and no other RAS effector binding domains have been demonstrated to oligomerize. A separate model of MLL-AF6 involvement in cellular transformation contends that loss of AF6 from membrane-proximal regions augments RAS signaling to alternative growth and differentiation effector pathways, leading to RAS-mediated cellular transformation[23]. However, whether wild-type AF6 truly functions as a RAS effector is not known.

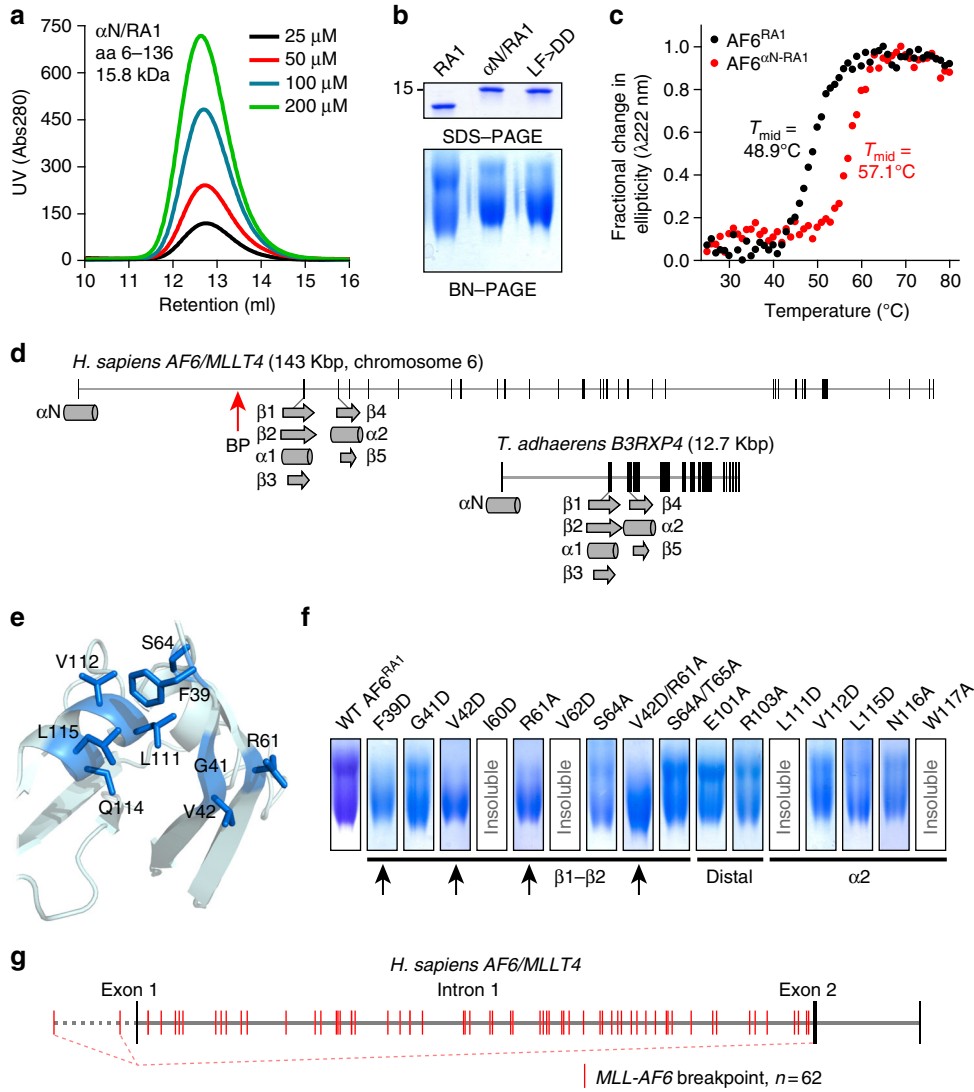

**Fig. 3** Inclusion of the αN helix stabilizes the AF6 RA1 domain and prevents its dimerization, but this secondary structure element is lost in all *MLL-AF6* patients. **a** Size exclusion chromatography of purified AF6 αN-RA1 domain (residues 6–136) presents a single monomer peak at several concentrations. **b** BN–PAGE analysis of purified core AF6 RA1 domain (37–136), αN-extended RA1 domain (6–136), and αN-extended RA1 domain double mutant deficient in RAS binding (L27D/F28D). **c** Thermal stability of the core (37–136, black) and αN-extended (6–136, red) RA1 domains. Melting curves were normalized to the maximal change in ellipticity monitored by far-UV CD at 222 nm. **d** Analysis of the human *AF6* gene structure shows that exon 1 encodes the αN helix, is followed by a large intron 1, and exons 2 and 3 encode the remaining structural elements of RA1 (top). This gene structure is completely conserved in *B3RXP4* of the primordial metazoan *T. adhaerens* (bottom). The single known breakpoint with *MLL* resides in the first intron (red arrow). **e** Ribbons diagram displaying normally buried residues, mostly hydrophobic, from the core RA1 domain (blue, sticks) that are exposed upon loss of the αN helix. **f** Validation of the dimerization interface by systematic mutation of residues exposed by loss of the αN helix. BN–PAGE gels exhibit collapse of typical monomer-dimer fractions (as for wild-type, far left) to a single monomeric band if a mutation affects dimerization (arrows). Potential dimer interface residues in β1, β2, and α2 (identified in **e**) were mutated to Asp or Ala. Mutation of distal residues E101 and R103 had no effect. Several mutations resulted in insoluble protein (marked). **g** Position of breakpoints in *AF6* derived from 62 patients with the *MLL-AF6* translocation (red). 60 breakpoints were within intron 1. 2 breakpoints upstream of exon 1 result in an identical fusion protein due to a splicing event (dashed)

Here, we report the structure and mechanism behind AF6-RAS association and its dependence on a highly conserved, auxiliary α-helix. We demonstrate that AF6 and RASSF1-6 are the only RAS effectors with RA domains comprising this helix, which non-canonically recognizes the switch II region of activated RAS. These structure-function observations allowed elucidation of RA1 dimerization via a unique mechanism. Using this structural insight to an MLL dimer-inducer, we studied the prevalence and role of MLL dimerization, including identification of associated proteins, and show that inhibiting dimerization completely blocks leukemogenesis in mice.

## Results

**AF6 association with RAS**. We sought to determine whether AF6 is a true RAS effector and whether its RA1 domain is oligomeric. An evolutionary approach establishes AF6 amongst the most conserved RAS effectors. We performed BLAST analysis on predicted protein sequences from the primordial metazoan *Trichoplax adhaerens*, one of the earliest multicellular organisms[24], using 52 RA domains from the human proteome (Supplementary Fig. 1a). Sequence conservation and domain organization identified seven remarkably conserved RAS effectors including: RAF, PI3K, RASSF, RASIP, RALGEF, GRB, and AF6.

The RA1 domain of *T. adhaerens* B3RXP4 shares 66% identity with human AF6 RA1, and secondary structure predictions suggested a ββαββαβ ubiquitin fold characteristic of RA domains[25]. Based on these conserved properties, we designed an *E. coli* expression construct and purified the AF6 RA1 domain (residues 37–136). Following assignment of $^1$H/$^{15}$N backbone resonances, we could corroborate the predicted secondary structure elements using chemical shift indices (CSI; Supplementary Fig. 1b, c). Importantly, we observed two distinct RA1 fractions during purification by size exclusion chromatography (Fig. 1b), verified as monomer and dimer by multi-angle light scattering (MALS; Supplementary Fig. 2a). Isothermal titration calorimetry (ITC) was used to measure an affinity of AF6 RA1 for RAS-GMMPNP (Supplementary Fig. 2b). The determined $K_d$ of 17.8 μM is an order of magnitude weaker than most RA domains, accounting for the position of AF6 near the bottom of a RAS effector hierarchy[26]. These data establish AF6 as an effector, but do not explain its unusual propensity to dimerize.

To gain further insight to the structure and function of AF6 RA1 we solved a three dimensional structure of this domain complexed with RAS at 2.5 Å resolution (Fig. 1c and Supplementary Fig. 2c). RAS and AF6 crystallized with 1:1 stoichiometry, and RA1 adopted the predicted ubiquitin fold topology. Further, RA1 presented the same pattern of protein-protein interaction observed in other effectors, an inter-protein β strand between β2 of RA1 and the switch I region of RAS supported by an interaction with the C terminus of α1 in the RA domain (Fig. 1d). Coordination of the binding interface by a Lys residue in β2 is a conserved element of effector-RAS complexes, mediated by K58 in AF6. These structural data support our preliminary analysis and uphold AF6 as a RAS effector, but still could not resolve the low affinity of AF6-RAS binding or RA1 dimerization.

**Detection of an αN helix**. A search model used for refinement of the AF6-RAS structure was a complex between RASSF5/NORE1A and RAS (PDB 3DDC)[27]. Sequence alignments clustered AF6 RA1 with the domains of RASSF1-6 (Supplementary Fig. 1a). The RASSF5 RA domain exhibits the archetypal ubiquitin fold, but its structure revealed an additional α-helix N-terminal to the core domain that mediates a non-canonical interaction with the switch II region of RAS[27]. Superposition of our AF6-RAS structure with RASSF5-RAS demonstrates the position of this auxiliary helix (designated αN; Fig. 1e). Secondary structure predictions on residues N-terminal to the RA1 domain of AF6 suggested an analogous helix (Fig. 1f). To resolve this, we purified a RA1 extended 30 amino acids at the N-terminus (residues 6–136). CSI analysis based on assigned $^1$H/$^{15}$N backbone resonances for this extended domain confirmed a helical segment aligned to AF6 residues 6–20 (Fig. 2a and Supplementary Fig. 2d). Using these data, we built a structure of the AF6-RAS complex with incorporated αN helix by homology modeling (Fig. 2b). At the interface are two residues from AF6 αN (Leu27/Phe28, corresponding to Cys220/Leu221 of RASSF5) that mediate an interaction with RAS switch II residues Ile36, Tyr64, and Met67. The Leu-Phe motif is completely conserved in AF6 orthologues (Fig. 2c), and we took advantage of this to functionally validate our model. We postulated that an extended AF6 RA1 domain would have increased affinity to RAS via the αN-switch II interface, and that mutation of the Leu-Phe motif would decrease affinity. Indeed, using ITC we obtained a $K_d$ of 4.1 μM for αN-extended RA1, 4-fold tighter than the core RA1 domain (Fig. 2d and Supplementary Fig. 3a). Mutation of AF6 residues Leu–Phe to negatively charged aspartate completely disrupted the interaction (Supplementary Fig. 3b). These results support our structural model and functionally correlate αN with RAS binding.

They also explain the inability of the core AF6 RA1 to compete with other RAS effectors[26], as the canonical domain lacks the auxiliary α-helix required for a high affinity interaction.

To determine the distribution of αN in RAS effectors, we performed a comprehensive analysis of residues N-terminal to β1 in the 52 human RA domains. Secondary structure predictions determine that AF6 and the RA domains of RASSF1-6 are the only RAS effectors with αN (Supplementary Fig. 3c). RIN1 and ARAP have α-helices N-terminal to β1 that are components of preceding domains (VPS9 and GAP, respectively). PI3K domains have several short helices, evident in a PI3K-RAS structure[28], but these do not contact the RAS G domain. Interestingly, PI3K does interface with the RAS switch II region through a unique interaction with K234 in β2[28]. RAPH1 and RAIM have N-terminal helices that function as coiled coils, analogous to Lamellipodin[29]. Thus, we propose that AF6 and RASSF1-6 are the only RA domains that bind RAS via the non-canonical switch II interface using this αN helix.

To substantiate the position of αN we used two NMR approaches. First, a 31 residue fragment encompassing the α-helix (AF6 6–36, which displays α-helical propensity) was titrated into $^{15}$N-AF6 RA1. We observed exchange broadening of residues in $^1$H/$^{15}$N-HSQCs corresponding to sequential segments in the β1, β2, and α2 regions of RA1, overlapping completely with our αN structure model (Supplementary Fig. 4a–c). In a second approach, we assessed combined chemical shift differences (Δppm) of backbone $^1$H/$^{15}$N resonances from the core RA1 domain versus the αN-extended domain (Supplementary Fig. 4d), a direct evaluation of αN-mediated rearrangements in RA1. We observed substantial Δppm values generated by αN incorporated in *cis* that correspond exactly with CSPs induced by αN in *trans* (Fig. 2e). These data validate position of the αN-helix, and reveal how functional evolution of the helix imparts a distinct mode of RAS recognition dependent on a structural component not present in most other effectors.

**Insertion of αN blocks leukemogenesis**. The *MLL-AF6* translocation generates an MLL fusion with AF6 residues 36–1651, truncating only 35 residues that comprise the αN helix (Fig. 1a). In contrast to core RA1 (37–136; Fig. 1b), we observed just a single monomer fraction during purification of αN-extended RA1 (Fig. 3a). This suggested that loss of αN in MLL-AF6 may be responsible for its dimerization via RA1. To examine this, we employed blue native PAGE (BN–PAGE)[30] to resolve oligomeric states for RA1 and the αN-extended domain. At 150 μM these proteins present clearly distinct dimerization capacities (Fig. 3b). RA1 displays two uniform bands, monomer and dimer, while αN-RA1 was almost completely monomeric. αN-RA1 further displayed increased thermal stability (Fig. 3c). In the *AF6* gene αN is coded by exon 1, which is followed by a long intron 1, and the remaining domain is coded by exons 2 and 3 (Fig. 3d). This organization is completely conserved in *T. adhaerens*, substantiating the importance of αN to AF6 function by its robust evolutionary conservation. Thus, dimerization of MLL-AF6 results from loss of the AF6 αN helix, which has been necessarily retained to facilitate RAS binding.

We sought to determine the mechanism of RA1 dimerization resulting from truncation of αN. Structural analysis revealed several residues in RA1 would be exposed upon loss of αN (Fig. 3e), primarily hydrophobic residues at the αN-RA1 interface (Phe39, Gly41, Val42, Leu111, Val112, Leu115, and also positively charged Arg61). We hypothesized that exposure of these residues would generate a hydrophobic patch nucleating dimerization, and systematically mutated them to aspartate (37–136). BN–PAGE showed F39D or V42D mutations collapse distinctive monomer-

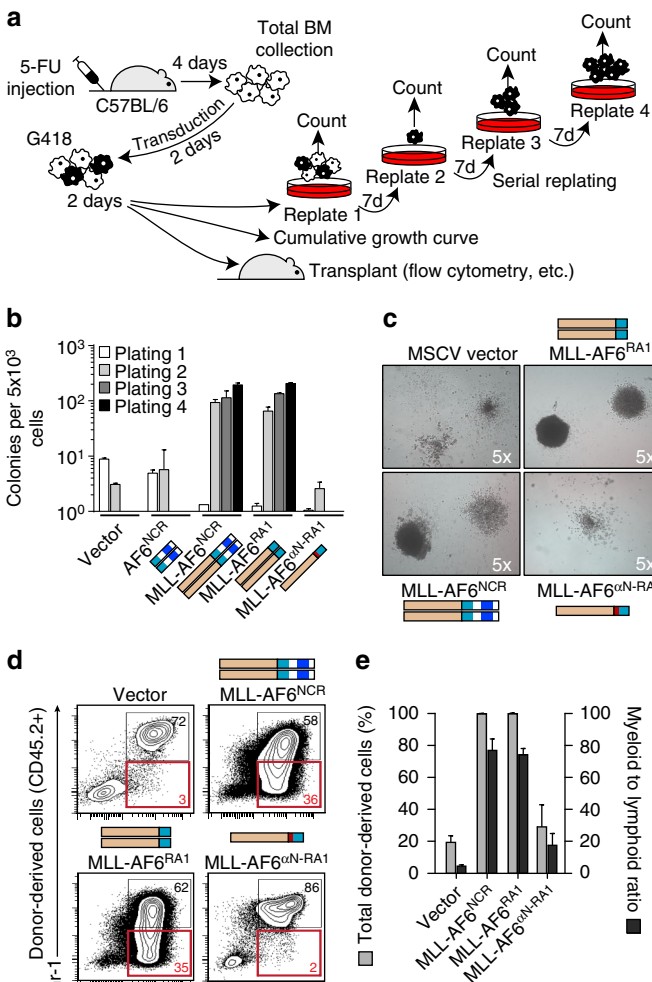

**Fig. 4** Insertion of the αN helix between MLL and AF6 disrupts myeloid immortalization and blocks its leukemogenic potential in mice. **a** Schematic of experimental strategies to measure immortalization of hematopoietic progenitors and induction of leukemogenesis. **b** Serial replating immortalization assay for progenitor cells transduced with retroviral constructs. Bars represent mean ± SD of the total colonies per 5 × 10³ cells derived from three replicates. **c** Typical granulocyte and macrophage colonies from primary methylcellulose cultures of pMSCVneo or MLL-AF6^αN-RA1 and macroscopic blast colonies from MLL-AF6^NCR or MLL-AF6^RA1 cultures. **d** Representative flow cytometry analysis of donor-derived myeloid cells in the bone marrow of transplanted mice. The red quadrants highlight the immature CD11b⁺GR1^low/negative myeloid subpopulation which was significantly expanded in MLL-AF6^NCR or MLL-AF6^RA1 expressing bone marrow cells in primary transplants compared to control cells (vector) and MLL-AF6^αN-RA1 expressing bone marrow cells. All mice transplanted with MLL-AF6^NCR or MLL-AF6^RA1 expressing cells were leukemic at 5 weeks ( > 20% blasts in the bone marrow and peripheral blood) while MLL-AF6^αN-RA1 expressing cells did not induce leukemia up to 10 weeks after transplantation. **e** Donor-derived engraftment in the bone marrow of mice transplanted with 10⁵ BM cells transduced with pMSCVneo, MLL-AF6^NCR, MLL-AF6^RA1 or MLL-AF6^αN-RA1. BM cells were selected for 7 days in G418 prior to transplantation. Shown are the mean ± SD of donor cell engraftment (CD45.2 + ) and of the myeloid (CD11b+, Gr1+) to lymphoid (CD3+, B220+) ratio in donor-derived cells 5 weeks after transplantation (2 ×10⁵/mice, n = 5 or 7 mice). Error bars represent s.d.

dimer bands to a single monomeric band (Fig. 3f). Mutations to distal sites or the α2-helix retained monomer–dimer equilibrium. While these data elucidate a mechanism for the single described *MLL-AF6* translocation[8], we considered whether this was archetypal. We determined *AF6* breakpoints using sequencing data from 62 patients possessing *MLL-AF6* translocations (Fig. 3g). 60 of these patients had breakpoints within intron 1. The two remaining breakpoints are spliced *MLL* partners[31] located upstream of *AF6* exon 1, which also generate fusions lacking only αN. Thus, all 62 patients express MLL-AF6 with only αN deleted, supporting a protein-selection determinant of leukemogenesis.

To directly address this, we used three approaches to assess whether re-insertion of αN between MLL and AF6 would disrupt its leukemogenic activity (Fig. 4a). Retroviral constructs were generated to express MLL fused to AF6 RA1 dimer (37–136) or αN/RA1 monomer (6–136). MLL-AF6^NCR (N-terminal conserved region, 35–348), equally leukemogenic as full length MLL-AF6[15], was used as a positive control while the vector and AF6^NCR alone (35–314) served as negative controls (Supplementary Fig. 5a–c). Cumulative growth curves of transduced primary mouse bone marrow (BM) cells revealed MLL-AF6^NCR and MLL-RA1 expressing cells in logarithmic growth, whereas MLL-αN/RA1 cells were indistinguishable from controls (Supplementary Fig. 5d). In serial replating assays, expression of MLL fused with AF6^NCR or RA1 conferred immortalization to myeloid progenitors while MLL-αN/RA1 did not (Fig. 4b). Morphologically, cells immortalized by either MLL-AF6^NCR or MLL-RA1 grew in compact blast colonies (CFU-Blast) while MLL-αN/RA1 resembled control cultures (Fig. 4c). The total numbers of CFU-Blasts expanded ~10⁹-fold over 3 replatings, indicating that dimerized MLL-AF6 drives a robust self-renewal. The expansion of CFU-blasts occurred at the expanse of multipotent progenitors (CFU-GEMM, Supplementary Fig. 5e) which were present in control cultures but absent from all experimental groups on first replating. Thus, introducing αN to block dimerization completely abrogated self-renewal activity. Finally, transduced BM cells were transplanted into 8-week to 12-week-old irradiated mice to determine oncogenic potential. All mice transplanted with MLL-AF6^NCR or MLL-RA1 were leukemic five weeks after transplantation. There was a 93–95% invasion of the bone marrow and peripheral blood by donor-derived cells which were exclusively myeloblasts (Supplementary Fig. 5g). The myeloid/lymphoid (M/L) ratio was 73–74 on average for MLL-AF6^NCR or MLL-RA1, compared to the expected M/L ratio of 5 in control groups. Therefore, our observations are consistent with the myeloid bias enforced by the oncogene, which was remarkably restored to near-normal by addition of αN. Moreover, re-insertion of αN brought reconstitution down to control levels observed with the empty vector (Fig. 4d). Mice receiving MLL-αN/RA1 cells were non-leukemic at 5 weeks and these mice remained leukemia-free out to 10 weeks. Finally, a hallmark of leukemic transformation is the capacity to re-induce AML in serial transplantation. We therefore re-transplanted MLL-AF6^NCR or MLL-RA1 expressing bone marrow cells into secondary and tertiary recipients (Supplementary Fig. 5f). In both secondary and tertiary transplantation, mice from both groups developed rapid and invasive myeloid leukemias that spread to the thymus and secondary lymphoid organs within one month (Fig. 4e and Supplementary Fig. 5h). Together, these results demonstrate that MLL-AF6-induced leukemogenesis critically depends on RA1 dimerization triggered by deletion of αN, a recurring structural determinant observed in all 62 MLL-AF6 translocations. This validates our biochemical data and confirms that truncation of the αN helix at the MLL-AF6 interface is critical for MLL activation.

**Interactors of AF6-induced MLL dimers**. Besides *AF6*, the five most common *MLL* translocation partners encode nuclear proteins involved in regulating gene expression (*AF4*, *AF9*, *AF10*, *ENL*, and *ELL*)[4]. Their fusion to MLL results in activation of transcription and epigenetic processes wholly consistent with their intrinsic biological functions, compartmentalization and interaction networks[18,32–35]. This includes interactions with three key regulatory complexes: DotCom[36], super elongation complex (SEC)[37] and the polycomb repressive complex (PRC)[38,39]. Association with WDR5 that mediates H3K4 methylation in complex with wild-type MLL[40,41] is not observed with MLL fusions, owing to loss of the C-terminal SET domain. It is not known if dimer-activated MLL invokes similar biochemical mechanisms to stimulate gene expression, as no MLL fusion with an oligomeric partner has been studied from a systems perspective. To resolve this, we sought to exploit the capacity for αN to inhibit dimerization and identify proteins interacting specifically with MLL dimers. As MLL fusions are large nuclear proteins, we choose not to use standard affinity-based proteomics but a proximity biotinylation assay called BioID[42]. MLL baits were fused with a mutant biotin ligase (BirA*) that covalently modifies preys in living cells and permits identification of proximal proteins. This method allows extraction of nuclear proteins with strong detergents and high salt concentrations without concerns of disrupting key interactions[43]. Our baits included the N-terminal region of MLL alone (MLLN), MLL-AF6 RA1 dimer (37–137), or MLL-AF6 αN/RA1 monomer (6–136). We filtered data from two biological replicates with SAINT[44], identifying 199 high-confidence bait-prey relationships (Supplementary Data 1). Gene ontology (GO) annotation of the global data set indicates MLLN, MLL-RA1 and MLL-αN/RA1 associate primarily with RNA processing and ribosomal proteins in the nucleolus (Fig. 5a), but also with transcription and chromatin modifying proteins in the nucleoplasm. This is consistent with a nuclear localization of MLL-AF6[5] and detection of other MLL partners at subnuclear foci[45]. As we were predominantly interested in monomer/dimer-specific partners, we filtered preys biotinylated specifically by monomeric MLLN and MLL-αN/RA1, or by dimeric MLL-RA1 (Fig. 5b, c). Dimers exclusively associated with nucleoplasm proteins, whereas monomeric-exclusive partners were nucleolar. We enriched these monomer/dimer-specific interaction sets with data from iRefIndex[46] (Supplementary Data 2). The resulting network diagram revealed two clearly distinct interaction sets, one specific to monomeric MLLN and MLL-αN/RA1, and another to dimeric MLL-RA1 (Fig. 6a). Significantly, the MLL dimer network contains components of two complexes previously implicated in MLL gene regulation: SEC and DotCom. The SEC complex was previously detected at sites of MLL-AF6 gene activation in the absence of physical interactions[47]. The central component of DotCom, the DOT1L histone methyltransferase, is known to play a vital role in MLL-AF6 leukemogenesis[48,49]. To validate interactions we performed immunoprecipitations of MLL fusion proteins followed by Western blotting for endogenous binding partners (Fig. 6b). All proteins encompassing the N-terminal half of MLL co-precipitated the tumor suppressor MENIN, known to directly bind MLLN[50,51] and uniformly identified by BioID, but only full length MLL-AF6 and MLL-RA1 bound DOT1L. Insertion of αN between MLL and AF6 completely abrogated this interaction. Under the stringent conditions necessary for nuclear extraction of these proteins (1% NP-40, 1% sodium deoxycholate, 10% glycerol, 300 mM NaCl) we did not observe robust co-precipitation of AF17, AF10, or ELL with MLL-AF6. However, using the BirA* tag as a proximity sensor we consistently observe that insertion of the αN helix results in significantly decreased levels of AF17, AF10, and ELL biotinylation compared with MENIN (Fig. 6c).

Collectively, our results establish that MLL shuttles between the nucleolus and nucleoplasm, and that dimerization of MLL imparts a capacity to interact with key mediators of gene expression.

**Cytoplasmic *MLL* fusion partners**. Extensive work has gone toward rationalizing partners in the *MLL* recombinome. While fusion to proteins capable of oligomerization can activate MLL[17,18,20,22,52], it is not known if this is a dominant mechanism that could explain the numerous infrequent translocations. We therefore used a bioinformatic approach to assess whether 56 MLL partners[4] had oligomerization potential. 20 of these were annotated as nuclear proteins, which we excluded due to potentially distinct mechanisms (i.e., direct interactions with gene expression modulators). We then performed predictions for coiled coils within the remaining 36 partners, 20 of which contain high-confidence coiled coil regions that are located C-terminal to the MLL breakpoint (Supplementary Table 1 and Supplementary Fig. 6a). Overall, MLL partners are 5-fold more likely to contain coiled coil regions than a set of randomly selected proteins (Supplementary Table 2). 9 of the 16 partners with no coiled coils instead have recognized oligomerization domains, and another five contain C-terminal SH3 domains with dimerization potential (Supplementary Table 3 and Supplementary Fig. 6b, c). Thus, 34/36 cytoplasmic MLL partner proteins are highly likely to oligomerize (Fig. 7a). MLL fusion with one remaining partner, BTBD18, truncates a modular domain and fuses MLL with a small helix-turn-helix comprising a hydrophobic patch (Supplementary Fig. 6a, b). We postulated that exposure of these residues may function analogously to AF6 RA1, and generated a fusion of BTBD18 residues 97–134 with a RAS binding domain from the effector ARAF that is intrinsically monomeric (Supplementary Fig. 6c). BN–PAGE presents two distinct fractions, indicative of a monomer-dimer equilibrium precipitated by addition of the BTBD18 residues (Supplementary Fig. 6d). Therefore, dimerization mediated by exposure of hydrophobic core residues is a novel and conserved MLL activation mechanism, joining oligomerization via coiled coils or modular domains to rationalize nearly all non-nuclear partners in the MLL recombinome.

There remains a question as to the extent that oligomerization partners contribute to higher order MLL association in vivo. The large size of MLL fusion proteins and their localization to the densely-packed nucleolar region makes this difficult to quantitate. Some evidence for this has come from co-precipitation of differentially tagged MLL fusion proteins[15,17,20]. To examine whether the AF6 RA1-mediated dimerization of MLL can be quantified in vivo, and if re-insertion of the αN helix can disrupt this, we tagged MLLN, full length MLL-AF6, MLL-RA1 and MLL-αN/RA1 with both FLAG and GFP and co-expressed the differentially tagged proteins. Following immunoprecipitation with anti-GFP we probed for the presence of FLAG-tagged protein to assess the capacity for each to self-assemble. Indeed, we observed co-precipitation of all MLL variants including the unfused MLLN region (Fig. 7b). This suggests that endogenous co-factors, perhaps MENIN[53] or LEDGF[54], can alone induce a higher order complex of MLL and that oligomerizing fusions such as AF6 either augment this via increased affinity/avidity, or that self-association of fusion partners imparts a conformational change upon the MLL N-terminal region. Further biophysical study of these large fusion proteins both in vitro and in vivo will be required to resolve this important question.

## Discussion

Cancer genome projects are generating extensive repositories of somatic gene mutations and complex chromosomal

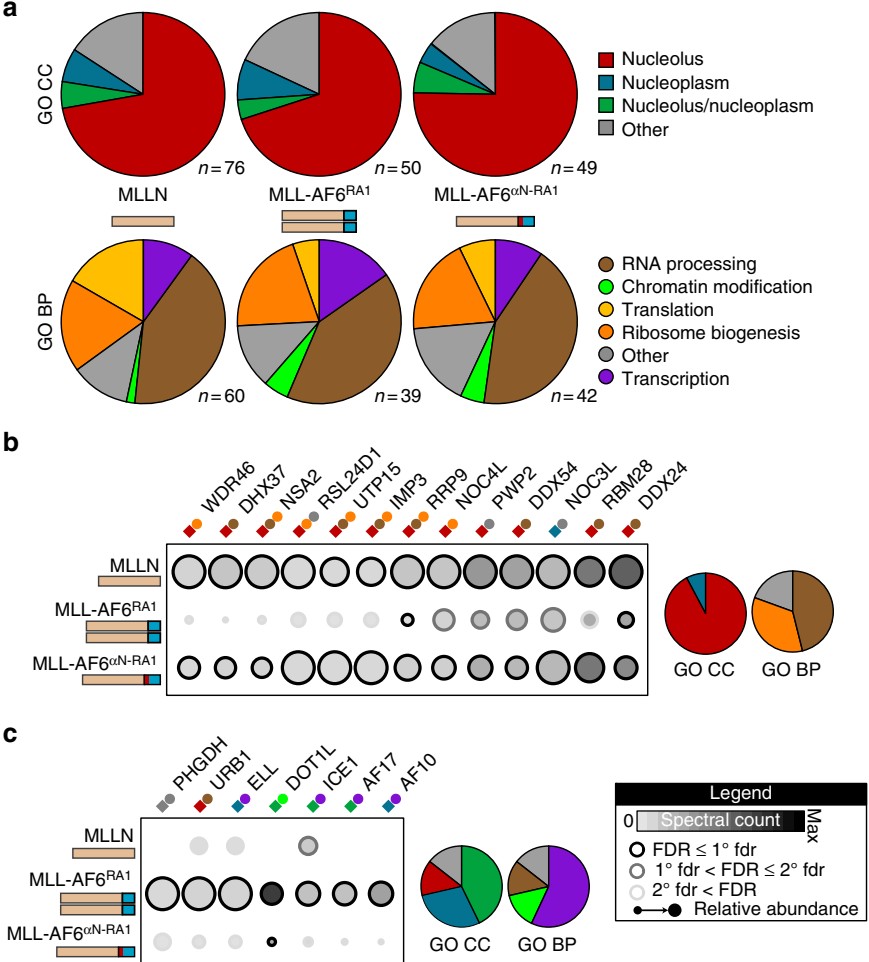

**Fig. 5** BioID proteomics reveal protein partners specific to monomeric and dimeric MLL-AF6 fusions. **a** Distribution of Gene Ontology (GO) cellular components (CC) denoting localization (top), and biological processes (BP) denoting protein function (bottom) of proteins associating with MLLN alone, the MLL-AF6 dimer (RA1, 37-136), or the MLL-AF6 monomer (αN-RA1, 6-136). n indicates the total number of identified biotinylated proteins following SAINT analysis. The majority of associating proteins were nucleolar (red) for all baits, though MLL-AF6[37-136] has a higher ratio of proteins localized to the nucleoplasm. **b** Proteins biotinylated specifically by MLLN and MLL-fused monomeric αN-RA1 domain. Annotation for GO CC (colored squares) and GO BP (colored circles) are from **a**. Chart summary of total CC and BP distributions are at right. **c** Proteins biotinylated specifically by the dimeric MLL-fused AF6 RA1 domain

translocations for numerous human cancers. To begin conceptualizing how these genetic abnormalities drive individual cancers, we must understand the complex molecular function of gene products. For *MLL*-driven leukemia's this has required comprehensive analysis of a myriad of seemingly random fusion partners. Here we provide a molecular mechanism of MLL activation resulting from its fusion with the RAS effector AF6, and evidence that oligomerization is a near universal property of MLL partners normally localized outside the nucleus, as has been speculated previously[17,18]. Together, these dimer-inducers constitute ~11% of *MLL* recombinations. Such gain-of-function fusions likely mimic oligomerization modulated intrinsically by the MLL PHD fingers[55], encoded immediately downstream of the *MLL* BCR and thus absent in fusion proteins. PHD-mediated oligomerization appears important to wild-type MLL epigenetic activity, but the PHD domains also provide targeting[56,57] and regulation[55], both lost in oncogenic fusions. From a systems standpoint, our functional proteomics data indicate that dimerized MLL recruits a network of protein complexes analogous to direct fusion of MLL with transcriptional activators. This suggests that targeting the epigenetic modulator DOT1L[48,49,58] should

have efficacy for all MLL-driven leukemia's. Conversely, an engineered MLL fusion with an FKBP domain capable of dimerization driven by a synthetic small molecule did not elicit robust H2K79 methylation[59], and may therefore not associate with DOT1L. To resolve weather all dimer-driven MLL fusions associate with DOT1L and induce H2K79 methylation will require a systematic analysis and improved biochemical understanding of the molecular complexes involved in chromatin modification and gene expression. Alternatively, we show that disruption of MLL-AF6 dimerization by re-insertion of αN completely inhibits leukemogenesis in a mouse model, and by extension individual dimer interfaces should serve as targets for molecular therapy. Remarkably, this holds true for two other leukemogenic fusion proteins: BCR-ABL[60] and PLZF/PML-RARα[61].

The combined genetic and protein functional mechanisms governing AF6-induced MLL activation exemplify the complex nature of chromosomal rearrangements that drive cellular transformation. αN plays a vital function in the association of AF6 with RAS, making it indispensable from an evolutionary standpoint. Between the αN coding exon and those encoding the

remaining RA1 domain is intron 1, which has expanded from 5.6 kb in *T. adhaerens* to 37.3 kb in humans and become a target for chromosomal rearrangement. Such intron expansion is characteristic of higher order genomes, and our data illustrate the potential deleterious effects this could have on fitness. Further, it is interesting to consider how frequently chromosomal translocations must occur to generate gain-of-function MLL fusion proteins with regularity, as only exposure of the hydrophobic residues at the $\alpha$N-RA1 interface provides transformative capacity in the context of MLL-AF6. *MLL* gene fusions occurring

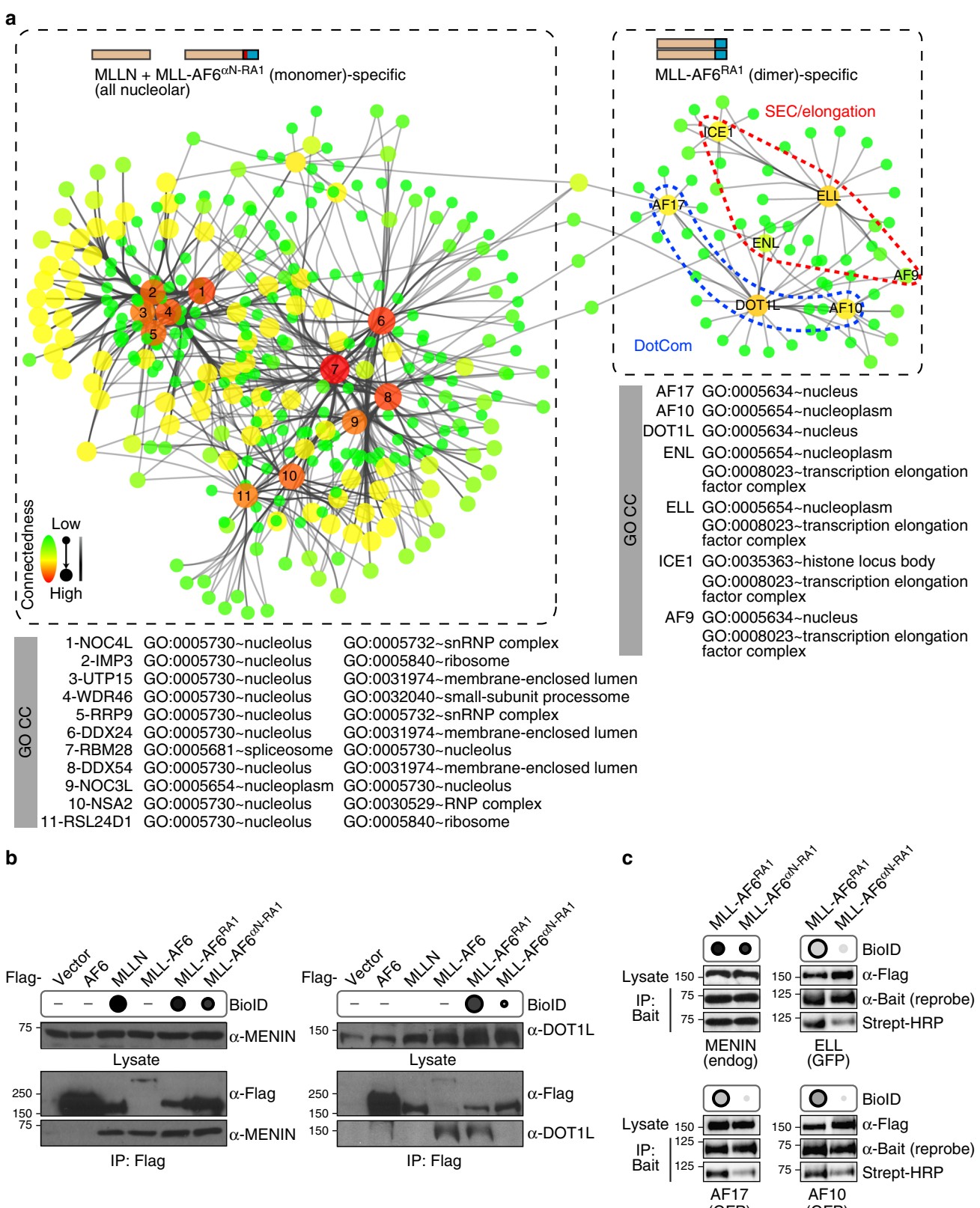

downstream of *AF6* intron 1, should they occur, would not confer growth advantages to target cells and are therefore not found in clinical samples. Our data thus shed light on an MLL activation mechanism, have implications for leukemogenic therapies, and identify a subset of RAS effectors which bind via a non-canonical mechanism.

## Methods

**Plasmid constructs and antibodies**. Human cDNA encoding wild-type H-RAS (Gene ID: 3265, residues 1–171) was cloned into pET15b (Novagen/EMD Biosciences) for bacterial expression with an N-terminal HIS tag. Constructs expressing RAS association domains from AF6 (Gene ID: 17356, residues 37–136 or 6–136) and ARAF (Gene ID: 369, residues 17–94) or the AF6 αN helix alone (6–36) with N-terminal glutathione *S*-transferase (GST) tags were sub-cloned into pGEX-4T2 (Amersham Pharmacia Biotech). To study dimerization by the BTBD18 fragment, cDNA encoding human BTBD18 (Gene ID: 643376, residues 97–134) was cloned in-frame, upstream of the ARAF RBD domain and expressed as a fusion from pGEX-4T2. For retroviral expression of MLL fusion proteins, MLL-AF6[NCR] (MLL residues 1–1395 fused to AF6 residues 35–348) and FLAG-AF6[NCR] (35–348) cloned into murine stem cell virus (MSCV) constructs were kindly provided by Michael Cleary[15]. AF6[NCR] was replaced with sequence encoding either the core AF6 RA domain (37-136) or the αN-extended RA domain (6–136). MLLN (residues 1-1395) was generated by excising AF6[NCR], blunt ending and ligation. For proteomic analysis of MLL-fused AF6 fragments, we generated BirA*/FLAG-tagged mammalian expression constructs by Gateway cloning into BirA*/FLAG pcDNA5 FRT/TO. These included MLLN (residues 1–1395), MLLN fused to AF6 RA domain (37–136), αN-RA domain (6–136), or full length AF6 (for co-immunoprecipitation experiments). These were shuttled to expression vectors with N-terminal FLAG and GFP tags for analysis of self-association. Mammalian expression vectors used for corroborating BioID results were also generated by Gateway cloning using cDNAs encoding full length AF17/MLLT6 (GeneID: 4302), AF10/MLLT10 (GeneID: 8028) and ELL (GeneID: 8178). All point mutations were performed by PCR-directed mutagenesis. All constructs were verified by sequencing. Monoclonal antibodies against FLAG M2 were purchased from Sigma (F1804; used at 1:5000), anti-DOT1L from Santa Cruz (C-3; 1:500), and anti-MLL from Millipore (05-764; 1:1000). Rabbit polyclonal anti-MENIN antibody was from Bethyl Laboratories (A300-105A; 1:1000) and anti-GFP from Abcam (ab290; 1:2500).

**Purification of recombinant proteins**. GST or HIS-tagged proteins were expressed in *E. coli* BL21 cells grown in minimal or LB media by induction with isopropyl-b-D-thiogalactopyranoside (IPTG) at 15 °C overnight. Generally, cells were lysed and sonicated in 20 mM Tris-HCl (pH 7.5), 150 mM NaCl, 10% glycerol, 0.4% NP-40, protease inhibitors (Roche), 1 mM phenylmethylsulfonyl fluoride (PMSF), 10 ng ml$^{-1}$ DNase, and either 1 mM dithiothreitol or 10 mM β-mercaptoethanol. Lysate was cleared by centrifugation and incubated with glutathione (Amersham Pharmacia Biotech) or Ni-NTA (Qiagen) resin at 4 °C for 1–2 h. Bound proteins were eluted directly with thrombin cleavage or with 250 mM imidazole (Bioshop) followed by thrombin. Concentrated proteins were purified to homogeneity by size exclusion chromatography using either an S75 or S200 column (GE Healthcare). Recombinant wild-type RAS was purified from *E. coli* predominantly in the GDP-bound form, and was loaded with GMPPNP[62].

**NMR spectroscopy**. NMR data were recorded at 25 °C on an 800 MHz Bruker AVANCE II spectrometer equipped with a 5 mm TCI CryoProbe, or a 600 MHz Bruker UltraShield spectrometer with 1.7 mm CryoProbe. NMR samples were prepared in buffer containing 20 mM Tris-HCl (pH 7.5), 100 mM NaCl, 1 mM DTT, 5 mM MgCl2 and 10% D$_2$O unless otherwise noted. Two-dimensional $^{1}$H/

$^{15}$N heteronuclear single quantum coherence (HSQC) spectra[63] as well as triple resonance HNCACB[64] and CBCACONH[65] spectra were collected for the backbone chemical shift assignments. Spectra were processed with NMRPipe[66] and analyzed using NMRView[67].

**Crystallography**. Crystals of AF6 RA1 (37-136) bound to RAS were obtained by incubating purified RA1 monomer fraction (450 μM) and GMPPNP-loaded RAS (450 μM) against a crystallization solution (20 mM K$_2$HPO$_4$, 18% (wt per vol) PEG-3350) by vapor diffusion at 25 °C. Crystals could not be obtained under the same conditions using αN/RA1 (6–136) and RAS. Crystals were soaked in crystallization solution containing 25% glycerol for data collection at 100 K. Diffraction data were collected at the Advanced Photon Source beamline 19 (Argonne National Laboratory, Argonne, IL) and processed with HKL2000[68]. Statistics pertaining to the diffraction data are in Supplementary Fig. 2c. The structure was determined at 2.5 Å resolution. The structure was solved by molecular replacement using the structure of RASSF5 RA domain complexed with RAS (PDB 3DDC)[27] as a search model. The final models were generated by successive rounds of refinement using PHENIX[69] accompanied by manual model building with Coot[70]. Modeling of the AF6 RA1 αN-helix was done using Coot and the RASSF5-RAS structure. Molecular graphics representations were prepared using PyMOL.

**Biochemical and biophysical protein analysis**. AF6 RA domain interactions with GMPPNP-loaded RAS were measured using a Microcal VP-ITC instrument. Stock solutions were diluted into filtered and degassed 20 mM Tris-HCl (pH 7.5), 100 mM NaCl, 5 mM MgCl$_2$ and 1 mM DTT. Experiments were carried out at 25 °C. Heats of dilution were determined from control experiments in which RA domains were titrated into buffer alone plus 500 μM GMPPNP. Data were fitted using the software Origin 7 (Microcal). Far-UV circular dichroism (CD) data were collected using a Jasco J-815 CD spectrometer (Jasco/Folio Instruments) with 0.1 cm path length ES quartz cuvettes, wavelength scan rates at 20 nm min$^{-1}$, a response time of 8 s and bandwidth of 1 nm. Spectra were corrected for buffer contributions. Thermal melts were acquired in 1 °C increments at a scan rate of 1 °C min$^{-1}$. Light scattering measurements were made with a Dynapro DLS module (Wyatt Technologies) using a scattering angle of 90° and incident laser light of 825 nm. BN–PAGE[30] was carried out using 15% acrylamide gels for separating RA domains. Proteins were purified by size exclusion chromatography and concentrated to 100–150 μM for BN–PAGE analysis.

**Cell culture and immunoprecipitation**. Human embryonic kidney epithelial (HEK 293T; ATCC CRL-3216) and human cervix epithelial (HeLa; ATCC CCL-2) cells were maintained in Dulbecco's Modified Eagle's Medium containing 10% fetal calf serum. For recombinant protein expression, cells were transiently transfected with PEI[71]. Stable HeLa cell lines for BioID analysis were generated as Flp-In T-REx cell pools (a kind gift from Arshad Desai, UCSD). For immunoprecipitation experiments, transfected cells were lysed in a modified RIPA buffer (50 mM Tris-HCl (pH 7.5), 1% NP-40, 1% sodium deoxycholate, 10% glycerol, 300 mM NaCl, 2 mM EDTA, 1 mM DTT, nuclease (benzonase, Sigma) and protease inhibitors). Following 1 h incubation to allow benzonase nuclease activity, lysates were cleared by centrifugation and incubated with pre-washed Protein G sepharose and immunoprecipitating antibody. Following 1–2 h incubation, beads were washed 5 times with modified RIPA, separated by SDS–PAGE and transferred to a nitrocellulose membrane for Western blot analysis. Membranes were blocked in TBST containing 5% (wt per vol) skim milk. Primary antibodies were detected with anti-mouse Ig or anti-rabbit Ig antibodies conjugated to horseradish peroxidase (1:10000) followed by treatment with ECL (Pierce). For detection of biotinylation, blots were probed with 1:10000 dilution of streptavidin-HRP (Sigma). Detection was done using either X-Ray film (ThermoFisher) and a film developer or with a Bio-Rad ChemiDoc imaging system equipped with the ImageLab software. All uncropped scans are provided in Supplementary Fig. 8.

**Fig. 6** Subcellular localizations and biological processes of dimer- or monomer-specific partner proteins. **a** Interaction network for proteins associating only with MLLN and the monomeric fusion MLL-AF6$^{αN-RA1}$, or only with the dimeric fusion MLL-AF6$^{RA1}$. Partners specific for dimer or monomer MLL proteins (Fig. 5b, c) were enriched with data from iRefIndex (specifically IntAct, BioGRID, MINT, and DIP) and imported to Cytoscape. Highly connected nodes (Betweenness Centrality) are larger and red, highly connected edges (Edge Betweenness) are opaque. Nodes less connected in the network are smaller and green, less connected edges have increasing transparency. Two inherently distinct clusters were established (dashed outlines) that reveal separate subcellular localization and protein functions. Monomeric MLL-specific interactors are nucleolar and are predominantly involved in ribosome biogenesis, while dimeric MLL-specific interactors are in the nucleoplasm and elicit transcriptional elongation. Two components of the MLL interactome are evident, the DotCom complex (blue) and the SEC complex (red). GO CC annotations for the key hub proteins at the core of these networks are listed at bottom. **b** Validation of BioID results by co-immunoprecipitation. HEK 293T cells were transfected with FLAG-tagged MLLN, MLL-fused dimeric AF6 RA1, or MLL-fused monomeric AF6 αN-RA1. FLAG-tagged wild-type AF6 (cytoplasmic) and MLL-fused full length AF6 were added, along with vector alone as controls. Dot representation of BioID results are at top. FLAG immunoprecipitations were probed for endogenous MEN1 (left) or DOT1L (right). **c** Validation of BioID results using biotinylation of specific associated proteins. MLL-fused to dimeric AF6 RA1 or monomeric αN-RA1 was co-expressed with GFP-tagged ELL, AF17 or AF10. anti-GFP immunoprecipitations were probed with streptavidin-HRP to quantify levels of biotinylation, and blots were re-probed with anti-GFP to observe loading. IP of endogenous MEN1, a constitutive binding partner of MLLN, was used as a control

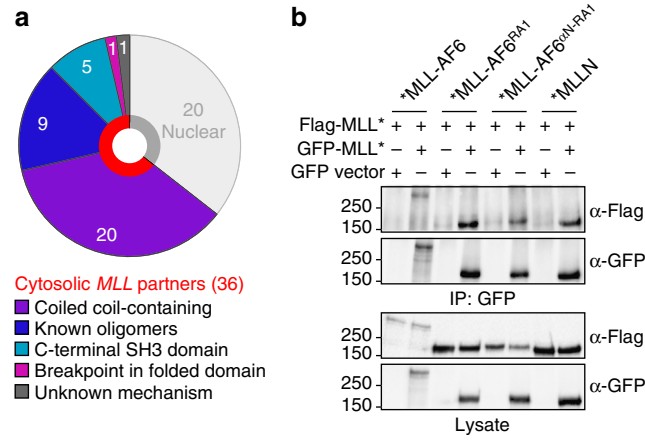

**Fig. 7** Dimerization potential of cytoplasmic MLL fusion proteins and the in vivo oligomerization of MLL. **a** 34 of 36 cytoplasmic partners in the *MLL* recombinome are known to dimerize, are predicted to have coiled coils, or have potentially dimerizing C-terminal SH3 domains. In addition, 20 of the partners are nuclear, one has no appreciated dimerization capacity, and another truncates a dimerization domain. **b** Co-precipitation of differentially tagged MLL proteins. FLAG- and GFP-tagged MLL-AF6, MLL-RA1, MLL-αN/ RA1 or MLLN alone were co-expressed in HEK 293T cells. Immunoprecipitation with anti-GFP followed by immunoblot with anti-FLAG determined potential for self-assembly. Co-expression with GFP alone served as a control

**In vivo oncogenic potential of MLL-AF6.** For gene transfer into bone marrow cells, pMSCV constructs consisting of in-frame fusions between MLL and various protein partners (MSCV, -FLAG-AF6$^{NCR}$, -MLL-AF6$^{NCR}$, -MLL-AF6$^{RA1}$, -MLL-AF6$^{αN-RA1}$) were used to produce high-titer, helper-free recombinant retroviruses by transduction of the mouse embryonic fibroblast retroviral packaging cell line GP + E86 (ATCC CRL-9642) with VSV-G-pseudotyped MSCV particles[72]. Twelve-week-old C57BL/6 mice were injected IP with 150 mg kg$^{-1}$ 5-Fluorouracil (5-FU; Sigma) and bone marrow cells were collected 4 days later for gene transfer. Briefly, cells were resuspended at 10$^6$ cells ml$^{-1}$ in Iscove modified Dulbecco's medium (IMDM; Gibco) supplemented with 15% (vol per vol) fetal bovine serum (FBS), 100 ng ml$^{-1}$ stem cell factor (SCF), 100 ng ml$^{-1}$ IL-11, 10 ng ml$^{-1}$ IL-6, 10 ng ml$^{-1}$ thrombopoietin and 50 mM α-monothioglycerol (Sigma). Cells were then co-cultured on irradiated (1500 cGy) virus-producing GP + E86 cells in the presence of polybrene (0.8 µg ml$^{-1}$; Sigma-Aldrich) for 48 h[73]. RT-PCR analysis to confirm expression was performed by extracting whole RNA (Qiagen) and generating cDNA with reverse transcriptase (Invitrogen) and oligo(dT) (Invitrogen). PCR was done using a forward primer specific for sequence encoding the C-terminus of MLL, and reverse primers specific for sequence encoding the AF6 αN (for MLL-αN/RA1), the AF6 RA1 domain (for MLL-RA1) or the AF6 RA2 domain (for MLL-AF6$^{NCR}$). For FLAG-AF6$^{NCR}$, the same reverse primer was used with a forward primer in pMSCV. For the transplantation assays, 8-week to 12-week-old recipient B6.SJK-*Ptprc$^a$ Pep3$^b$*/BoyJ (CD45.1$^+$) mice were irradiated at 800 cGy and transplanted with 2 × 10$^5$ retrovirally infected total BM (CD45.2$^+$) cells in conjunction with a life-sparing dose of host (CD45.1$^+$) total BM (5 × 10$^4$) cells[73,74]. Engraftment (donor-derived CD45.2$^+$ cells) was assessed by flow cytometry analysis of peripheral blood, bone marrow, spleen and thymus 5 weeks after transplantation using lineage specific antibodies: Gr-1, CD11b (myeloid cells), B220 (B cells), Thy1.2 (T cells), and CD71 (erythroid precursors). All animals were maintained in pathogen-free conditions according to institutional animal care and guidelines set by the Canadian Council on Animal Care. Permission for the animal experiments described here was granted by the University of Montréal Deontology Committee on Animal Experimentation.

**Methylcellulose colony-forming assays.** Retrovirally infected bone marrow cells were plated in duplicates at two concentrations (5 × 10$^4$ or 2 × 10$^5$ cells per ml) in methylcellulose medium (IMDM (Gibco), 10% fetal bovine serum (FBS), 1% methylcellulose (Fluka), 2% bovine serum albumin (BSA), 100 ng ml$^{-1}$ stem cell factor (SCF), 50 mM α-monothioglycerol (Sigma), 200 µg ml$^{-1}$ plasma transferrin, 1 U ml$^{-1}$ recombinant erythropoietin, and IL-3), with or without neomycin (G148; Gibco) 1 mg ml$^{-1}$ for 7–10 days. Neomycin-resistant colonies were scored and classified[73]. For serial replating assays, colony cells were collected, pooled and plated at 5 × 10$^3$ cells ml$^{-1}$ under the same conditions but without further neomycin selection. Secondary colonies were scored and subjected to tertiary and quaternary replating assays. Colonies were scored using a Leitz Labovert inverted microscope (Leitz Wetzlar).

**Flow cytometry analysis.** Following selection with neomycin (G148) at 1 mg ml$^{-1}$ for 72 hours, Kit + Sca + Lin- (KSL) and progenitor population FACS analysis was performed[74] using BD Pharmigen and eBioscience antibodies for c-Kit (CD177; 2B8), Sca-1 (Ly-6A/E; D7), CD16/32 (93), CD34 (RAM34), IL7Ra (CD127; A7R34), and Lin + cells were excluded by staining with biotinylated antibodies against B220 (CD45R; RA3-6B2), CD11b (M1/70), GR1 (Ly-6C/G; RB6-8C5), CD3e (145-2C11), Thy1.2 (CD90.2; 30-H12). Lineage-positive population FACS analysis was also performed using BD Pharmigen and eBioscience antibodies for B220 (CD45R; RA3-6B2), CD11b (M1/70), GR1 (Ly-6C/G; RB6-8C5), CD3e (145-2C11), Thy1.2 (CD90.2; 30-H12). Dead cells were excluded by propidium iodide staining. FACS analysis was performed on an LSRII cytometer.

**Proteomics.** BioID[42,75] was done using HeLa Flp-In T-REx cells stably expressing BirA*/FLAG-tagged MLL fusion proteins grown in 2 × 150 cm$^2$ plates of sub-confluent cells (60%) incubated 24 h in complete media supplemented with 1 µg ml$^{-1}$ tetracycline (BioShop) and 50 µM biotin (BioBasic). Cell pellets were resuspended by pipetting up and down and vortexing in 1.5 ml of RIPA buffer (50 mM Tris-HCl pH 7.5, 150 mM NaCl, 1% NP-40, 1 mM EDTA, 1 mM EGTA, 0.1% SDS, protease inhibitors (Sigma), and 0.5% sodium deoxycholate). 1 µl of benzonase (250U) was added to each sample and the lysates sonicated on ice. Lysates were centrifuged for 20 min at 12,000 × g, and then incubated with streptavidin-sepharose beads (GE) pre-washed with RIPA buffer. Affinity purification was performed at 4 °C on a nutator for 3 h, beads were pelleted (400 × g, 1 min), the supernatant removed, and the beads washed 3 times in 1 ml RIPA buffer followed by 3 times in 1 ml 50 mM ammonium bicarbonate pH 8.0 (ABC). Residual ABC was removed and beads were resuspended in 100 µl of 50 mM ABC for protein digestion. 10 µl of a 0.1 µg µl$^{-1}$ trypsin stock (resuspended in 20 mM Tris-HCl, pH 8) was added for a final concentration of 1 µg of trypsin and incubated at 37 °C overnight. The following day, an additional 1 µg of trypsin was added (in 10 µl of 20 mM Tris-HCl, pH 8.0) and the samples incubated an additional 2–4 h. Beads were pelleted (400 × g, 2 min) and the supernatant (peptides) transferred to a fresh 1.5 ml tube. Beads were rinsed 2 times in 100 µl HPLC water and pooled with the collected supernatant. Formic acid was added to a final concentration of 2% to end digestion (30 µl of 50% stock). The pooled supernatant was then centrifuged at 10000 × g for 10 min and the supernatant collected and lyophilized. Peptides were resuspended in 5% formic acid and one quarter of the sample was analyzed per MS run. 5 µl of each sample was directly loaded at 400 nl min$^{-1}$ onto a 75 µm × 12 cm emitter packed with 3 µm ReproSil-Pur C$_{18}$-AQ (Dr. Maisch HPLC GmbH, Germany). The peptides were eluted from the column over a 90 min gradient generated by a NanoLC-Ultra 1D plus (Eksigent, Dublin CA) nano-pump and analyzed on a TripleTOF 5600 instrument (AB SCIEX, Concord, Ontario, Canada). The gradient was delivered at 200 nl min$^{-1}$ starting from 2% acetonitrile with 0.1% formic acid to 35% acetonitrile with 0.1% formic acid over 90 min followed by a 15 min clean-up at 80% acetonitrile with 0.1% formic acid, and a 15 min re-equilibration period in 2% acetonitrile with 0.1% formic acid for a total of 120 min. To minimize carryover between each sample, the analytical column was washed for 3 hours by running an alternating sawtooth gradient from 35% acetonitrile with 0.1% formic acid to 80% acetonitrile with 0.1% formic acid, holding each gradient concentration for 5 min. Analytical column and instrument performance were verified after each sample by loading 30 fmol BSA tryptic peptide standard (Michrom Bioresources, Inc., Fremont, CA) with 60 fmol α-Casein tryptic digest and running a short 30 min gradient. TOF MS calibration was performed on BSA reference ions before running the next sample in order to adjust for mass drift and verify peak intensity. The instrument method was set to a discovery or IDA mode which consisted of 1 250 ms MS1 TOF survey scan from 400–1300 Da followed by 20 100 ms MS2 candidate ion scans from 100–2000 Da in high sensitivity mode. Only ions with a charge of 2+ to 4+ which exceeded a threshold of 200 cps were selected for MS2, and former precursors were excluded for 10 s after 1 occurrence.

MS data generated by TripleTOF 5600 were stored, searched and analyzed using the ProHits laboratory information management system (LIMS) platform[76]. Within ProHits, the resulting WIFF files were first converted to an MGF format using WIFF2MGF converter and to an mzML format using ProteoWizard[77] (v3.0.4468) and the AB SCIEX MS Data Converter (V1.3 beta) and then searched using Mascot (v2.3.02) and Comet (v2012.02 rev.0). The spectra were searched with the human and adenovirus complements of the RefSeq database (version 57) from NCBI supplemented with "common contaminants" from the Max Planck Institute (http://141.61.102.106:8080/share.cgi?ssid = 0f2gfuB) and the Global Proteome Machine (GPM; http://www.thegpm.org/crap/index.html). Parameters included: fully tryptic cleavages, allowing up to 2 missed cleavage sites per peptide. The mass tolerance was 40 ppm for precursors with charges of 1+ to 3+ and a tolerance of ±0.15 amu for fragment ions. Variable modifications were deamidated asparagine and glutamine and oxidized methionine. The results from each search engine were analyzed through TPP (the Trans-Proteomic Pipeline[78], v4.6 OCCUPY rev 3) via the iProphet pipeline[79]. Two unique peptides ions and a minimum iProphet probability of 0.95 were required for protein identification prior to analysis with SAINTexpress version 3.3[44]. Eight control runs were used for comparative purposes: 4 runs of a BioID analysis conducted on cells expressing the BirA*/FLAG tag only to control for non-specific biotinylation of intracellular proteins, and 4 runs from a BioID analysis conducted on an unrelated bait protein

(GFP) to mimic the condition in which endogenous biotinylation (which primarily occurs on mitochondrial carboxylases) would be predominant. Each negative control was analyzed in biological replicates with four independent biological replicates per type of control (i.e., not simple re-injections or technical replicates). A compression strategy using SAINTexpress collapsed the 8 controls to the highest 4 spectral counts for each hit, helping to capture spurious binding behavior of some contaminants. Thus, each potential prey across the two biological replicates of the bait is assessed for significance across the four highest values across the eight controls we used. Only proteins passing a statistical threshold of FDR ≤0.02 were deemed high quality interactions and are reported in Supplementary Data 1. For subsequent analysis of preys specific to monomeric MLLN and MLL-αN/RA1, or to dimeric MLL-RA1, we used only preys passing a statistical threshold of FDR ≤0.01 (Fig. 5, Fig. 6, Supplementary Data 2).

**Data availability**. Coordinates and structure factors for the AF6-RAS crystal structure are deposited in the protein data bank (PDB) with accession code 6AMB. The mass spectrometry data has been deposited as a complete submission in the ProteomeXchange through partner MassIVE (massive.ucsd.edu), and assigned identifiers PXD007631 and MSV000081499, respectively. The data that support the findings of this study are available from the corresponding authors upon reasonable request.

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

## Acknowledgements

This work was supported by grants from the Cancer Research Society (CRS; to M.I., T.H., and A.-C.G.), the Canadian Cancer Society Research Institute (CCSRI; to M.I. and T.H.), the Canadian Institutes for Health Research (CIHR; to M.J.S., M.I., T.H., and A.-C.G (FDN 143301)), the Princess Margaret Hospital Foundation (to M.I.), the Leukemia Lymphoma Society (LLS) of Canada (to T.H.), the National Science and Engineering Council of Canada (NSERC; to T.H.), and the MERST Québec and Fonds de Recherche du Quebec (FRQ)-Nature et Technologie (to T.H.). DFG-Grant MA1876/11-1 supported the contribution of C.M. and R.M., and E.O. received a CIHR studentship. Canada Research Chairs are held by M.I. in Cancer Structural Biology, A.-C.G. in Functional Proteomics and M.J.S. in Cancer Signaling and Structural Biology. The infrastructure of IRIC is supported in part by a group grant from FRQ-Santé. The Canada Foundation for Innovation funded the 800- and 600-MHz NMR spectrometers and the mass spectrometers.

## Author contributions

This project was conceived and designed by M.J.S and M.I. M.J.S., E.O., N.I., M.G., A.H., S.M., and R.C.L. performed experiments and analyzed data. M.T. and G.G.-S. supplied technical support and experimental assistance. C.M. and R.M. assembled and analyzed genomic sequencing data. M.D.M., A.-C.G. and T.H. supervised experimental designs and analyzed data. The manuscript was prepared by M.J.S and M.I., with input from N.I., M.D.M., A.-C.G. and T.H.

## Additional information

**Competing interests:** The authors declare no competing financial interests.

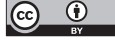

