## [Peer Review File · Nature Communications]

Reviewers' comments:

Reviewer #1 (Remarks to the Author): Expert in MLL biochemistry

The authors employed a structural biology approach to demonstrate that the RAS effector protein AF6 contains a unique alpha helix upstream of the RAS-interaction domain (RA1) that is critical for non-canonical RAS interaction. Of note, this helix modulates the ability of RA1 to oligomerize and its consistent absence in MLL-AF6 fusion proteins is critical for their oncogenic potential presumably by unmasking latent oligomerization. The studies provide an elegant protein structural illumination for RAS association by its AF6 effector. However, the oncogenic implications are not entirely novel since previous studies have correlated the leukemogenic potential of MLL-AF6 with oligomerization potential, and MLL-AF6 has been shown to interact on chromatin with SEC. Furthermore, the pathogenic importance of oligomerization has been demonstrated for several MLL fusion proteins. The unique aspect of the current study lies with the protein structural insights it provides. The manuscript could be improved by focusing on the novel aspects, primarily structural data, and deleting much of the speculative parts dealing with breakpoint locations and evolutionary implications. Most importantly, the authors need to appropriately place their structural observations in context of numerous previous studies characterizing the mechanism of MLL-AF6 leukemogenesis and the already established role of MLL dimerization in general.

1. In the Introduction, the authors do not accurately portray observations of previous studies. For example, the authors state that “It has been proposed that AF6-mediated MLL activation is dependent on its first RAS association domain, and that this domain might self-associate”. However, studies in Ref 15 actually demonstrated that AF6 mediated MLL activation dependent on RA1 and that this domain did indeed self-associate. Similarly, this same study showed that RAS binding domain can oligomerize in contrast to the authors’ statement that “no data indicating that RAS binding domains have any capacity to oligomerize”. Also, the previous study provided mechanistic understanding of MLL-AF6 in contrast to the statement that “There remains no mechanistic understanding of MLL-AF6”. Finally, the authors’ do not acknowledge that previous studies have demonstrated that MLL-AF6 recruits SEC components to chromatin of target genes as part of a unifying mechanism for MLL fusion partners.

2. The last section of the manuscript (MLL Breakpoint Positions, Figure 7, Discussion last paragraph) is very speculative (and more of a literature review as opposed to primary data) and

should be deleted to narrow the focus.

3. It is unclear what conclusion can be drawn from the proteomic study. The association of MLLN, MLL-RA1 and MLL- α N/RA1 primarily with RNA processing and ribosomal proteins may be due to incomplete MS data analysis. Common background proteins such as biotinylated carboxylases, ribosomal proteins should be removed from the list.

4. Figure 3b: The mutant LF>DD displays less dimer compared to α N/RA1 indicating these two amino acids are also critical for dimerization. What is BN-PAGE analysis of LF>DD in RA1 only?

5. Fig 4b and Sup Fig 5d,e lack statistical analyses.

6. Affinity of AF6 RA1 and RAS-GMMPNP should be in Sup Fig 2b.

Reviewer #2 (Remarks to the Author): Expert in structural studies and NMR

This paper describes some very nice work involving a wide variety of structural, proteomic and functional studies to understand the impact of a translocation involving the MLL and AF6 proteins. I think it is definitely worthy of publication but I have some reservations relating in particular to the way results are described and the way the conclusions are drawn.

My main problem is that I do not understand the molecular mechanism that the authors are proposing for MLL activation resulting from its fusion with AF6. The biochemical and structural studies provide strong evidence that loss of the helix to the N-terminus of the RAS binding domain leads to a propensity of the region of the AF6 protein that becomes fused to MLL to dimerise. However, the authors also mention previous work (Fair et al., *Mol. Cell Biol.*, 2001, 21: 3589–3597) that shows that PHD domains in the C terminal half of MLL protein also mediate homo-dimerisation. In the MLL-AF6 translocation the C terminal half of MLL, which mediates homo-dimerization, is replaced with most of AF6, which lacks the N terminal helix and also provides a dimerisation function. Why do the authors think that replacing the intrinsic dimerisation domain(s) in MLL with a different dimerisation domain from AF6 provides “a molecular mechanism for MLL activation resulting from its fusion with AF6”? This doesn’t make sense to me. (I also note that the enzymatic SET domain in MLL is lost in the translocation.) Could the MLL-AF6 fusion act as a dominant negative that prevents MLL activity and does that lead to transformation? Do they really mean “a molecular mechanism for AF6 activation resulting from homo-dimerisation and its aberrant localisation to the nucleus via MLL”? I think a much clearer description and justification of the mechanism they propose and

perhaps some experiments to test this are needed here.

Previous work had established that the AF6 RAS-binding domain self associates and that it is sufficient to transform cells when fused to MLL (Liedtke et al., *Blood*, 2010, 116: 63-70). In this work the authors determined the structure of the RAS/AF6 RAS-binding domain complex, and show that it is very similar to the structure of the RAS/RASSF5 complex solved previously (Stieglitz et al., *EMBO J.* 2008, 27: 1995-2005). Initially, they studied an AF6 RAS-binding domain lacking the N terminal α -helix, but this bound weakly to RAS. They later realised that they needed the α N-helix (as seen in the RAS/RASSF5 structure) in order to get a high affinity interaction. Interestingly, in the course of these experiments, they realised that the RAS-binding domain from AF6 in the absence of this N-terminal helix has a propensity to dimerise, whereas when the N terminal helix is present it forms a monomer. In a nice set of experiments, they then went on to show that when a MLL-AF6 fusion protein is made that includes the α N-helix the protein is non-transforming.

Based on these results, they suggest that MLL-AF6 induced leukemogenesis critically depends on RAS-binding domain dimerization, triggered by the deletion of this α N-helix. However, I think this is an assumption. It could be that the removal of this α N-helix exposes a hydrophobic patch, which leads to the AF6 part of the fusion protein having a propensity to dimerise (although this is not demonstrated *in vivo*). However, it seems equally plausible to me that exposure of the sticky hydrophobic patch resulting from α N-helix removal could also lead to aberrant association with other proteins and that might underlie its transforming activity.

The authors then went on to carry out proteomic experiments to look at interactions with the MLL-AF6 fusion. They initially characterized the MLL interactome using available proteomic data. However, the figures provided here are very confusing with WDR5 indicated as a complex, whereas I think it is mainly a component of the several different MLL complexes. Why not show it as such? There also appear to be other complexes identified in Supplementary Figure 6a (which are not mentioned). I think this Figure requires more work. I also found Figure 6a very confusing and it is hard to work out how what interactions they identify for the different bait proteins. I suspect that their proteomics analysis is made more complex by the fact that they are using a proximity biotinylation assay, which identifies a very large number of interactions with nearby proteins that the MLL-AF6 fusion may or may not bind to directly. They say they chose not to use standard affinity based proteomics (was there any reason?), but given that these assays are carried out in living cells where substantial chromatin dynamics may allow the enzyme to modify many nearby proteins I was not sure how suitable this assay is. I think a much clearer presentation is needed here, but nevertheless (as I mention above) it is not clear to me why they think that dimerization of MLL imparts a capacity to interact with key mediators of gene expression, when the region that has been removed from MLL in the translocation is thought to homo-dimerise anyway?

In summary, the authors uncover the fact that loss of the α N-helix of the RAS-binding domain in AF6 during the translocation has a key impact on the transforming activity of the fusion. They provide bioinformatics analysis to suggest that other MLL partners may have oligomerization tendencies and experimental data to show that a small helix-turn-helix from the BTBD18 protein causes a different RAS binding domain from ARAF to also have a propensity for dimerization. The results therefore suggest that dimerization may be a conserved MLL activation mechanism, but no data is presented to suggest that the MLL protein dimerises in vivo to any greater extent when fused to AF6 than it does by itself, and therefore that MLL dimerization is the key factor underlying leukemogenesis.

Smaller points:

In general I think there are too many statements, which don't quite seem right. For example:

(1) They mention that AF6 and RASF1-6 are the only RAS-binding domains that bind RAS via "the non-canonical switch II interface". However, my impression is that many RAS effectors also interact with and stabilise the dynamics of the switch II interface.

(2) They mention that 34/36 MLL partner proteins can oligomerize. However, since most of this is established through bioinformatics studies, I think it would be more appropriate to say 'may oligomerize'.

(3) They mention that dimerization mediated by exposure of hydrophobic core residues is a novel and conserved MLL activation mechanism. However, again, I think it might be better to suggest that it 'may be' a conserved mechanism.

(4) They mention that they demonstrate that MLL-AF6 physically complexes with DOT1L. However, given that they are using a proximity biotinylation assay, they cannot possibly determine from this experiment that it "physically complexes" with DOT1L.

In conclusion, whilst I think this paper includes some very nice work, for someone like me who has not worked in the RAS/cancer field for a long time it does need to be written a lot more carefully so that the reader can understand the results and follow the story.

Rebuttal

Reviewer #1: Expert in MLL biochemistry

We thank the reviewer for recognizing “*our studies provide an elegant protein structural illumination for RAS association by its AF6 effector*” and have tried to satisfy their request to simplify the manuscript in this revised version. The major critique from this reviewer was that “*previous studies have correlated the leukemogenic potential of MLL-AF6 with oligomerization*” and that “*the pathogenic importance of oligomerization has been demonstrated for several MLL fusion proteins*”. This is encapsulated in the reviewer’s first point:

1. In the Introduction, the authors do not accurately portray observations of previous studies. For example, the authors state that “It has been proposed that AF6-mediated MLL activation is dependent on its first RAS association domain, and that this domain might self-associate”. However, studies in Ref 15 actually demonstrated that AF6 mediated MLL activation dependent on RA1 and that this domain did indeed self-associate. Similarly, this same study showed that RAS binding domain can oligomerize in contrast to the authors’ statement that “no data indicating that RAS binding domains have any capacity to oligomerize”. Also, the previous study provided mechanistic understanding of MLL-AF6 in contrast to the statement that “There remains no mechanistic understanding of MLL-AF6”. Finally, the authors’ do not acknowledge that previous studies have demonstrated that MLL-AF6 recruits SEC components to chromatin of target genes as part of a unifying mechanism for MLL fusion partners.

The reviewer provides two points here which we must dispute. First, on the mechanistic insights provided by previous work: The first evidence that dimerization can activate MLL came serendipitously, *via* observation that *Mll-lacZ* caused leukemia in mice¹. Subsequent work demonstrated that inducible dimers can cause leukemia², and that GAS7^{3,4}, AF1p/EPS15⁴, SEPT6⁵, EEN/SH3GL1⁶ and GPHN⁷ have predicted dimerization motifs which can activate MLL activity. Thus, there are five primary reports that dimerization of MLL may activate its epigenetic activity, plus work from Cleary’s group on the more common AF6 fusion⁸ (studied here). Those works summarized:

Partner	Dimerization	Method	ITC/AUC/NMR	Protein Structure
GAS7 ^{3,4}	coiled-coil	GST pull down/co-IP	No	No (None available)
EPS15 ⁴	coiled-coil	co-IP	No	No (None available)
SEPT6 ⁵	coiled-coil	co-IP	No	No (None available)
EEN ⁶	coiled-coil	GST pull down/co-IP	No	No (BAR available ⁹ , not cc)
GPHN ⁷	15 aa region	cross-linking	No	No (None available)
AF6 ⁸	RA domain	co-IP	No	No (None available)

You will note there is not a single solved structure of an oligomerization domain from any MLL fusion partner. There have been NO biophysical experiments performed with ANY ‘dimerizing’ MLL fusion partner (ITC, NMR or AUC). In previous work on AF6 by Liedtke *et al.*⁸ there is not a single *in vitro* expressed protein or any biophysical analysis of this interaction at all. Thus, when we state that “There remains no mechanistic understanding of MLL-AF6” and that there

are “no data indicating that RAS binding domains have any capacity to oligomerize”, this is accurate. We would assert that co-immunoprecipitation of differentially tagged proteins from cell lysates is not a definitive biophysical approach to demonstrate protein-protein dimerization. Further, the mechanism was clearly not deduced as there is no elucidation of the dimer interface or any of the key residues mediating oligomerization; and the AF6 dimer is very unique compared to the coiled coils of other partners (as our work demonstrates). In actuality, these types of molecular analyses are lacking for many MLL fusions and also the MLL protein itself. Thus, while we acknowledge that oligomerization is a recognized avenue through which MLL can be activated, molecular mechanisms remain absent. Further, there is no evidence that dimerization is a universal property responsible for the wide variation in partner genes, and no systematic study of protein networks associated with MLL dimers for target ID (as we did with BioID). As evidence that these are open and key questions, several quotes from recent reviews:

Robert Slany: “*Unfortunately, up to now it is not known how dimerized MLL fusion proteins activate target genes.*”¹⁰ -2009

Ali Shilatifard *et al.*: “*how MLL N-terminal dimerization drives progenitor cell self-renewal is unclear.*”¹¹ -2010

Jay Hess and Andrew Muntean: “*The mechanisms by which the dimerization of truncated MLL makes it transforming are unknown.*”¹² -2012

Akihiko Yokoyama: “*the molecular mechanism by which the dimerization of MLL fusion proteins causes transformation remains elusive.*”¹³ -2015

To address the reviewers concerns we have added the complete list of previously studied MLL oligomer partners (as above) to the introduction, and more accurately detailed what was known previously about MLL-AF6 (introduction, paragraph 2). The second point on which we must disagree is that “*MLL-AF6 recruits SEC components to chromatin of target genes as part of a unifying mechanism for MLL fusion partners.*” If indeed this were true it may provide a unifying mechanism for MLL fusion partners, but we find no evidence of this in the literature. In fact, it has been demonstrated that MLL-AF6 does NOT co-precipitate with the SEC components AF4, AF5q31, cyclin T1, ENL or CDK9¹⁴. Instead, these components are recruited to sites of MLL-AF6 activated transcription “*via unknown alternative mechanisms*”¹⁴. Thus, this does not suggest a unifying mechanism but rather that distinct MLL fusions require conserved transcriptional machinery that can be recruited by diverse processes. In addition to these SEC components, it has also never been demonstrated that MLL-AF6 can directly complex with DOT1L, and MLL-AF6 has been excluded from all previous proteomic analyses. The reviewers point that SEC components are present at chromatin of MLL-AF6 target genes is an important one, and we’ve added this information to the Results section (subsection ‘Interactors of AF6-Induced MLL Dimers’).

2. *The last section of the manuscript (MLL Breakpoint Positions, Figure 7, Discussion last paragraph) is very speculative (and more of a literature review as opposed to primary data) and should be deleted to narrow the focus.*

To narrow the focus of our manuscript, as suggested by the reviewer, we have removed the speculative parts in the last paragraph and also the results and discussion pertaining to what was Figure 7a/b/c (also removed). To further simplify we also removed Supplemental Figure 6a/b and Supplemental Tables 1 and 4. The remains of Figure 7 (previously e and f) were moved to the new Supplemental Figure 7 and are now only mentioned in the text (Results).

3. It is unclear what conclusion can be drawn from the proteomic study. The association of MLLN, MLL-RA1 and MLL- α N/RA1 primarily with RNA processing and ribosomal proteins may be due to incomplete MS data analysis. Common background proteins such as biotinylated carboxylases, ribosomal proteins should be removed from the list.

Our MS data analysis was in fact extremely stringent and we can be very confident that the reported partners are derived exclusively from our bait proteins. Common background components are removed from our data sets in several ways. To summarize, we used eight control runs including four runs on cells expressing BirA/Flag and another four on an unrelated bait. The proteins listed passed an extremely high statistical threshold of FDR (false discovery rate) ≤ 0.01 , and this is following removal of common contaminants and also iProphet and SAINTexpress calculations to determine the probability of each hit from background contaminants. The full process was in the Materials and Methods:

“The spectra were searched with the human and adenovirus complements of the RefSeq database (version 57) from NCBI supplemented with “common contaminants” from the Max Planck Institute (<http://141.61.102.106:8080/share.cgi?ssid=0f2gfuB>) and the Global Proteome Machine (GPM; <http://www.thegpm.org/crap/index.html>). Parameters included: fully tryptic cleavages, allowing up to 2 missed cleavage sites per peptide. The mass tolerance was 40 ppm for precursors with charges of 1+ to 3+ and a tolerance of ± 0.15 amu for fragment ions. Variable modifications were deamidated asparagine and glutamine and oxidized methionine. The results from each search engine were analyzed through TPP (the Trans-Proteomic Pipeline¹⁵, v4.6 OCCUPY rev 3) via the iProphet pipeline¹⁶. SAINTexpress version 3.3¹⁷ was used as a statistical tool to calculate the probability value of each potential protein-protein interaction from background contaminants using default parameters. Two unique peptide ions and a minimum iProphet probability of 0.95 were required for protein identification prior to SAINTexpress analysis. Eight control runs were used for comparative purposes; 4 runs of a BioID analysis conducted on cells expressing the BirA*/Flag tag only, and 4 runs from a BioID analysis conducted on an unrelated bait protein (GFP). The 8 controls were collapsed to the highest four spectral counts for each hit. Only proteins passing a statistical threshold of $FDR \leq 0.02$ were deemed high quality interactions and are reported in Supplementary Table 2. For subsequent analysis of preys specific to monomeric MLLN and MLL- α N/RA1, or to dimeric MLL-RA1, we used only preys passing a statistical threshold of $FDR \leq 0.01$ ”

Thus, we can actually be very clear about our conclusions from the proteomic study as we are using a cut-off with extremely high stringency, and further got hits that make complete sense from an MLL signaling perspective. More information on this can be found in these references¹⁶⁻¹⁹. This is in fact the first work to provide a structure/function analysis of a fused protein triggering MLL dimerization, allowing us to study MLL dimers from a systems standpoint. The BioID²⁰ approach was essential as large, nuclear MLL proteins are difficult to

extract and purify – but this method allows use of harsh detergents and high salt concentrations because we are not concerned with disrupting protein-protein interactions. BioID has not previously been applied to study MLL. Our data clearly indicate that dimerized MLL shuttles from the nucleolus to sites of active transcription, where it associates with gene expression modulators (specific components of the DOTcom and SEC complexes). It was alleged that MLL-AF6 did not associate with these proteins¹⁴, and no other MLL dimers have been studied. Microscopy work has previously detected MLL in nuclear ‘speckles’^{21,22}, but our BioID approach truly demonstrates the dynamic behaviour of activated MLL translocating from the nucleolus (implying a novel therapeutic strategy). We also spent five months refining a reproducible technique and repeating endogenous co-IP experiments, as we believed MLL dimer association with DOT1L of enormous general interest and never previously shown. AF6 is cytoplasmic and MLL-AF6 is a 340 kDa nuclear protein, and here we are probing MLL fusion associated with endogenous, low-level nuclear proteins (not over-expressed proteins) and believe our experiments of exceptional quality considering the numerous complexities we had to overcome.

4. *Figure 3b: The mutant LF>DD displays less dimer compared to Δ N/RA1 indicating these two amino acids are also critical for dimerization. What is BN-PAGE analysis of LF>DD in RA1 only?*

The Leu-Phe motif (mutated to Asp-Asp) is at the bottom of the α N helix and not present in the core RA domain.

5. *Fig 4b and Sup Fig 5d,e lack statistical analyses.*

These have been added.

6. *Affinity of AF6 RA1 and RAS-GMMPNP should be in Sup Fig 2b.*

The affinity is listed at the bottom and is also stated in the text.

Reviewer #2 (Remarks to the Author): Expert in structural studies and NMR

We again thank the reviewer for recognizing our “*very nice work involving a wide variety of structural, proteomic and functional studies to understand the impact of a translocation involving the MLL and AF6 proteins*” and for deeming “*it is definitely worthy of publication*”. We address here the concerns of the reviewer:

1) *I do not understand the molecular mechanism that the authors are proposing for MLL activation resulting from its fusion with AF6. The biochemical and structural studies provide strong evidence that loss of the helix to the N-terminus of the RAS binding domain leads to a propensity of the region of the AF6 protein that becomes fused to MLL to dimerise. However, the authors also mention previous work (Fair et al., Mol. Cell Biol., 2001, 21: 3589–3597) that shows that PHD domains in the C terminal half of MLL protein also mediate homo-dimerisation. In the MLL-AF6 translocation the C terminal half of MLL, which mediates homo-dimerization, is replaced with most of AF6, which lacks the N terminal helix and also*

provides a dimerisation function. Why do the authors think that replacing the intrinsic dimerisation domain(s) in MLL with a different dimerisation domain from AF6 provides “a molecular mechanism for MLL activation resulting from its fusion with AF6”? This doesn’t make sense to me. (I also note that the enzymatic SET domain in MLL is lost in the translocation.) Could the MLL-AF6 fusion act as a dominant negative that prevents MLL activity and does that lead to transformation? Do they really mean “a molecular mechanism for AF6 activation resulting from homo-dimerisation and its aberrant localisation to the nucleus via MLL”? ... It seems equally plausible to me that exposure of the sticky hydrophobic patch resulting from α N-helix removal could also lead to aberrant association with other proteins and that might underlie its transforming activity.

The reviewer has asked for clarification on our proposed molecular mechanism – i.e. that constitutive dimerization through the AF6 RA1 domain can mimic an activating dimerization mediated by the PHD domains of wild-type MLL. This requires some knowledge of previous work done on the MLL protein – an extensive and sometimes contradictory body of literature that we will attempt to summarize as minimally as possible. Full length MLL is post-translationally cleaved (C-terminal to the PHD domains) to generate N- and C-terminal fragments that may interact with each other, may increase protein stability and may govern nuclear localization²³ though some have disputed this²⁴. The key feature here is that N-terminal MLL retains the complete PHD repeat region, and is functional for transcriptional activation. The PHD fingers of MLL have been shown to co-precipitate²⁵, but importantly also regulate MLL activity *via* interactions with histones^{26,27} and other nuclear proteins²⁵. Ultimately, which genes are expressed by MLL epigenetic activity is therefore controlled by the PHD region, and this influence must be lost in MLL fusions²⁸ (i.e. they are mutually exclusive). Not surprisingly then, the MLL breakpoint cluster region is immediately upstream of the PHD fingers. We propose that an MLL fusion partner such as AF6 can mediate a homodimerization, analogous to the PHD fingers, but the regulatory activity imparted by the wild-type MLL PHD domains is lost and the epigenetic activity becomes constitutive. We have included a more complete clarification of this model in our discussion at the reviewer’s request.

2) *The authors then went on to carry out proteomic experiments to look at interactions with the MLL-AF6 fusion. They initially characterized the MLL interactome using available proteomic data. However, the figures provided here are very confusing with WDR5 indicated as a complex, whereas I think it is mainly a component of the several different MLL complexes. Why not show it as such? There also appear to be other complexes identified in Supplementary Figure 6a (which are not mentioned). I think this Figure requires more work. I also found Figure 6a very confusing and it is hard to work out how what interactions they identify for the different bait proteins. I suspect that their proteomics analysis is made more complex by the fact that they are using a proximity biotinylation assay, which identifies a very large number of interactions with nearby proteins that the MLL-AF6 fusion may or may not bind to directly. They say they chose not to use standard affinity based proteomics (was there any reason?), but given that these assays are carried out in living cells where substantial chromatin dynamics may allow the enzyme to modify many nearby proteins I was not sure how suitable this assay is. I think a much clearer presentation is needed here, but nevertheless (as I mention above) it is not clear to me why they think that dimerization of*

MLL imparts a capacity to interact with key mediators of gene expression, when the region that has been removed from MLL in the translocation is thought to homo-dimerise anyway?

The reviewer is asking for clarification of our MLL interactome (i.e. regarding some of the known MLL interactors from previous proteomics studies). We thought this information useful as a comparison for our own proteomic data, as no MLL-dimer partner had previously been studied by MS/BioID. First, WDR5 is not a ‘*component of several different MLL complexes*’ but an interacting protein that binds the SET domain of wild-type MLL (therefore lost in MLL fusions). It was conceivable that MLL fusions could bind to proteins within this complex, reproducing what is seen with the wild-type protein. Our results, however, found components only of the SEC and DOTCOM complexes. To satisfy the reviewer, we’ve actually removed Supplemental Figure 6a as it was a compendium of previous results and perhaps caused some confusion. As for the network figures we are presenting, these are by far the most commonly and used and effective method to depict multiple bait-prey interactions in modern experimental proteomics, and we have tried to improve the clarity of Figure 6a. Contrary to the reviewers comments we are actually showing very few interactions due to the high stringency (confidence) cut-offs we used in our proteomic analysis (see also response to reviewer 1, point 3). The answer to the question on our choice of BioID is in the text: “*As MLL fusions are large nuclear proteins, we choose not to use standard affinity-based proteomics but a proximity biotinylation assay called BioID. MLL baits were fused with a mutant biotin ligase (BirA*) that covalently modifies preys in living cells and permits identification of proximal proteins.*” Immunoprecipitation of these large nuclear proteins is difficult and doing it with intact binding partners is highly impracticable, as their extraction from the nucleus requires harsh detergents and high salt that disrupts these interactions. We added this point to the text. We believe BioID will become the method of choice to study network interactions for all nuclear proteins in the near future, and are happy to be at the forefront of that. Finally, “*dimerization of MLL imparts a capacity to interact with key mediators of gene expression*” is an accurate statement regardless of whether wild-type MLL can homo-dimerize or not. Indeed, this statement reflects the fact that MLL fusions may very well mimic properties of the wild-type protein, but in a constitutive fashion (see also our response to reviewer 2, point 1).

3) *The authors uncover the fact that loss of the α N-helix of the RAS-binding domain in AF6 during the translocation has a key impact on the transforming activity of the fusion. They provide bioinformatics analysis to suggest that other MLL partners may have oligomerization tendencies and experimental data to show that a small helix-turn-helix from the BTBD18 protein causes a different RAS binding domain from ARAF to also have a propensity for dimerization. The results therefore suggest that dimerization may be a conserved MLL activation mechanism, but no data is presented to suggest that the MLL protein dimerises in vivo to any greater extent when fused to AF6 than it does by itself, and therefore that MLL dimerization is the key factor underlying leukemogenesis.*

As this reviewer is an expert in structural studies and NMR, they could attest to the fact that there are no available approaches whereby we could study the 340 kDa MLL-AF6 protein from a biophysical standpoint. Michael Cleary’s group has shown that differentially tagged MLL-AF6 proteins can co-precipitate, and this requires the RA domain⁸, but that is the extent to which we can study the full protein with available technologies. In fact, this is what makes our molecular

understanding of the dimer interface so important – it allowed the first description of this MLL dimer vs a monomer by comparisons with MLL-AF6 in which we re-inserted the α N helix to prevent dimerization. The mouse studies and BioID proteomic studies we present are the first to compare monomer and dimer *in vivo*.

4) *They mention that AF6 and RASFI-6 are the only RAS-binding domains that bind RAS via “the non-canonical switch II interface”. However, my impression is that many RAS effectors also interact with and stabilise the dynamics of the switch II interface.*

This is not correct, as most RAS effectors bind the RAS switch I region (or effector binding region) as stated in text. RASSF5 was the only known effector to interact with RAS switch II before our study. Our AF6-RAS structure will be just the 7th RAS-effector complex structure, remarkable considering the level of interest in this field. We describe how the AF6 and RASSF proteins 1-6 share a non-canonical binding mode to RAS that is dependent on the auxiliary N-terminal α -helix. Thus, just 7/54 RBD domains in the human proteome (Figure S3c) have a capacity to interact with switch II, noteworthy when we consider downstream effectors are currently the most reliable therapeutic targets in RAS-driven cancers (RAF and PI3K).

5) *They mention that 34/36 MLL partner proteins can oligomerize. However, since most of this is established through bioinformatics studies, I think it would be more appropriate to say ‘may oligomerize’.*

We have changed this in the text.

6) *They mention that dimerization mediated by exposure of hydrophobic core residues is a novel and conserved MLL activation mechanism. However, again, I think it might be better to suggest that it ‘may be’ a conserved mechanism.*

While we did demonstrate that the BTBD18 fusion oligomerizes *via* an analogous mechanism, to recognize the reviewer’s concern we have changed the wording regarding a universal conservation of this mechanism.

7) *They mention that they demonstrate that MLL-AF6 physically complexes with DOT1L. However, given that they are using a proximity biotinylation assay, they cannot possibly determine from this experiment that it “physically complexes” with DOT1L.*

Perhaps the reviewer has missed our demonstration in Figure 6b that MLL-AF6 and DOT1L can co-immunoprecipitate? These co-IPs were done to confirm some of the interesting BioID results, and we spent five months refining a reproducible technique and repeating endogenous co-IP experiments, as we believed MLL dimer association with DOT1L of enormous general interest and never previously shown. Again, MLL-AF6 is a 340 kDa nuclear protein and cannot be 100% extracted from the nucleus but we believe our experiments of exceptional quality considering the numerous complexities we had to overcome. Indeed, our data answers questions put forth by Cleary’s group⁸ when they suggested “*MLL-AF6 does not form hybrid complexes (with AF4 and ENL family proteins designated AEP), but nevertheless constitutively recruits AEP to target chromatin via unknown alternative mechanisms.*” We show here that AF6 does indeed physically

complex with DOT1L and AF17 (Figure 6b), the first demonstration that an MLL dimer does so and thus an observation with clinical relevance. An inhibitor of DOT1L methyltransferase activity is currently in Phase I clinical trial (EPZ-5676). Unfortunately, this does not mean that MLL leukemias have been cured. In fact, initial reports from Epizyme and Celgene conclude favourable dose response in 30 patients with recombined MLL-r leukemias, but remission in just 2 patients (6.7%). Dr. Scott Armstrong (Director, Center for Epigenetics Research and Leukemia Center at Sloan Kettering) confirmed these numbers at a recent seminar (November, 2015), revealing that patients who demonstrated an initial biological response (loss of MLL-stimulated gene expression) relapsed via re-establishment of HOX gene expression and showed resistance to DOT1L inhibition. Thus, these cancers will undoubtedly require combinatorial therapies and new target ID, making inhibition of dimerization hugely significant, in addition to our identification of ELL, ICE1, AF17 and AF10 as dimer-specific protein partners.

References

1. Dobson CL, Warren AJ, Pannell R, Forster A, Rabbitts TH. Tumorigenesis in mice with a fusion of the leukaemia oncogene Mll and the bacterial lacZ gene. *EMBO J.* 2000;19(5):843–851.
2. Martin ME, Milne T a, Bloyer S, et al. Dimerization of MLL fusion proteins immortalizes hematopoietic cells. *Cancer Cell.* 2003;4(3):197–207.
3. So CW, Karsunky H, Passegué E, et al. MLL-GAS7 transforms multipotent hematopoietic progenitors and induces mixed lineage leukemias in mice. *Cancer Cell.* 2003;3(February):161–171.
4. So CW, Lin M, Ayton PM, Chen EH, Cleary ML. Dimerization contributes to oncogenic activation of MLL chimeras in acute leukemias. *Cancer Cell.* 2003;4(2):99–110.
5. Ono R, Nakajima H, Ozaki K, et al. Dimerization of MLL fusion proteins and FLT3 activation synergize to induce multiple-lineage leukemogenesis. *J. Clin. Invest.* 2005;115(4):919–929.
6. Liu H, Chen B, Xiong H, et al. Functional contribution of EEN to leukemogenic transformation by MLL-EEN fusion protein. *Oncogene.* 2004;23(19):3385–94.
7. Eguchi M, Eguchi-Ishimae M, Greaves M. The small oligomerization domain of gephyrin converts MLL to an oncogene. *Blood.* 2004;103(10):3876–82.
8. Liedtke M, Ayton PM, Somerville TCP, Smith KS, Cleary ML. Self-association mediated by the Ras association 1 domain of AF6 activates the oncogenic potential of MLL-AF6. *Blood.* 2010;116(1):63–70.
9. Masuda M, Takeda S, Sone M, et al. Endophilin BAR domain drives membrane curvature by two newly identified structure-based mechanisms. *EMBO J.* 2006;25(12):2889–2897.
10. Slany RK. The molecular biology of mixed lineage leukemia. *Haematologica.* 2009;94(7):984–93.
11. Mohan M, Lin C, Guest E, Shilatifard A. Licensed to elongate: a molecular mechanism for MLL-based leukaemogenesis. *Nat. Rev. Cancer.* 2010;10(10):721–8.
12. Muntean AG, Hess JL. The pathogenesis of mixed-lineage leukemia. *Annu. Rev. Pathol.* 2012;7:283–301.
13. Yokoyama A. Molecular mechanisms of MLL-associated leukemia. *Int. J. Hematol.* 2015;101(4):352–361.
14. Yokoyama A, Lin M, Naresh A, Kitabayashi I, Cleary ML. A higher-order complex containing AF4 and ENL family proteins with P-TEFb facilitates oncogenic and physiologic MLL-dependent transcription. *Cancer Cell.* 2010;17(2):198–212.
15. Pedrioli PG. Trans-proteomic pipeline: a pipeline for proteomic analysis. *Methods Mol Biol.* 2010;604:213–238.
16. Shteynberg D, Deutsch EW, Lam H, et al. iProphet: Multi-level Integrative Analysis of Shotgun Proteomic Data Improves Peptide and Protein Identification Rates and Error Estimates. *Mol. Cell. Proteomics.* 2011;10(12):M111.007690-M111.007690.
17. Teo G, Liu G, Zhang J, et al. SAINTexpress: Improvements and additional features in Significance Analysis of INTeractome software. *J. Proteomics.* 2014;100:37–43.
18. Choi H, Liu G, Mellacheruvu D, et al. Analyzing protein-protein interactions from affinity purification-mass spectrometry data with SAINT. *Curr. Protoc. Bioinforma.* 2012;(SUPPL.39):1–31.
19. Lambert J, Tucholska M, Go C, Knight JDR, Gingras A. ScienceDirect Proximity

- biotinylation and affinity purification are complementary approaches for the interactome mapping of chromatin-associated protein complexes ☆. *J. Proteomics*. 2014;118:81–94.
20. Roux KJ, Kim DI, Raida M, Burke B. A promiscuous biotin ligase fusion protein identifies proximal and interacting proteins in mammalian cells. *J. Cell Biol.* 2012;196(6):801–10.
 21. Zeisig DT, Bittner CB, Zeisig BB, et al. The eleven-nineteen-leukemia protein ENL connects nuclear MLL fusion partners with chromatin. *Oncogene*. 2005;24(35):5525–32.
 22. Caslini C, Alarcón a S, Hess JL, et al. The amino terminus targets the mixed lineage leukemia (MLL) protein to the nucleolus, nuclear matrix and mitotic chromosomal scaffolds. *Leukemia*. 2000;14(11):1898–908.
 23. Hsieh JJ, Ernst P, Tempst P, et al. Proteolytic Cleavage of MLL Generates a Complex of N- and C-Terminal Fragments That Confers Protein Stability and Subnuclear Localization. *Mol. Cell. Biol.* 2003;23(1):.
 24. Yokoyama A, Ficara F, Murphy MJ, et al. MLL becomes functional through intramolecular interaction not by proteolytic processing. *PLoS One*. 2013;8(9):e73649.
 25. Fair K, Anderson M, Bulanova E, et al. Protein Interactions of the MLL PHD Fingers Modulate MLL Target Gene Regulation in Human Cells. *Mol. Cell. Biol.* 2001;21(10):.
 26. Milne T a, Kim J, Wang GG, et al. Multiple interactions recruit MLL1 and MLL1 fusion proteins to the HOXA9 locus in leukemogenesis. *Mol. Cell*. 2010;38(6):853–63.
 27. Chang P-Y, Hom R a, Musselman C a, et al. Binding of the MLL PHD3 finger to histone H3K4me3 is required for MLL-dependent gene transcription. *J. Mol. Biol.* 2010;400(2):137–44.
 28. Chen J, Santillan D a, Koonce M, et al. Loss of MLL PHD finger 3 is necessary for MLL-ENL-induced hematopoietic stem cell immortalization. *Cancer Res*. 2008;68(15):6199–207.

Reviewers' comments:

Reviewer #1 (Remarks to the Author):

The authors for the most part have addressed the comments raised in prior review. However, there is one point that has not been appropriately addressed. In the introduction, the authors state “It has been proposed that AF6-mediated MLL activation is dependent on its first RAS association (RA) domain (ref 15), and ...” The referenced study actually demonstrated that the RA1 domain was the minimal region sufficient for oncogenesis in structure/function experiments as encapsulated in the title of the publication “Self-association mediated by the Ras association 1 domain of AF6 activates the oncogenic potential of MLL-AF6”. Thus, the sentence needs to be revised to state “It has been demonstrated that AF6-mediated MLL activation is dependent on its first RAS association (RA) domain.....”.

Reviewer #2 (Remarks to the Author):

Thank you for sending me the revised version of the manuscript entitled “Evolution of AF6-RAS Association and its Implications in Mixed-Lineage Leukemia” Sadly, the authors have not addressed the questions raised from the first version of their manuscript, and I am still not convinced by the molecular mechanism that the authors are proposing for MLL activation resulting from its fusion with AF6.

They believe that they “provide evidence that oligomerisation is the dominant mechanism driving oncogenesis from rare MLL translocation partners”. Their mechanism may be correct, but whilst I think they do provide evidence that deletion of the alpha-N-helix in the Ras-binding of AF6 could lead to dimerisation of MLL-AF6, it doesn't appear to be a particularly tight interaction. It is therefore not clear whether MLL-AF6 exists predominantly as a dimer in vivo, and I do not think the results necessarily imply that it is dimerisation that leads to the recruitment of DOT1L etc. Deletion of the alpha-N-helix clearly results in the exposure of a sticky hydrophobic patch on the surface of the MLL-AF6 protein. This could easily result in the aberrant recruitment of other proteins or complexes, depending on the relative amounts of

monomer and dimer in vivo. Notably, even at high concentrations in vitro most of the AF6 RA1 domain protein is monomeric – see Figure 1b. Why do the authors think it is dimerisation of MLL-AF6 that leads to the recruitment of DOT1L etc.? I still think an equally plausible mechanism is that the N terminal domain of the MLL proteins in some way targets the protein to chromatin and its fusion with AF6 results in the aberrant recruitment of proteins such as DOT1L to MLL sites via this sticky patch, and that is what leads to transformation. I do not think the authors provide any clear evidence that “oligomerisation” is the dominant mechanism.

In their rebuttal the authors propose “that an MLL fusion partner such as AF6 can mediate homo-dimerization, analogous to the PHD fingers, but the regulatory activity imparted by the wild-type MLL PHD domains is lost and the epigenetic activity becomes constitutive”. This also does not make sense to me. Why do they think the epigenetic activity becomes constitutive when the enzymatic domain that mediates H3K4 methylation (the SET domain) is also lost in this fusion? What epigenetic activity are we talking about? It is possible that by reading the literature in this area in detail I could perhaps understand what is going on here, but I had hoped that the authors would come back (at least in the rebuttal) with a more detailed analysis of their data in the context of the existing literature and provide a solid argument of why they think their mechanism is preferred.

Regarding point 2, I do not want to get into an argument about the pros and cons of different proteomics methods, but I do think it important that they point out that there is a family of MLL complexes containing different MLL proteins. Maybe all the literature is wrong, but there seem to be many papers describing WDR5 as a component of different MLL complexes (see e.g. Hubert et al., *Epigenetics*. 2013, 8, 79-91). It is also not clear in the text/figures which MLL protein they are actually working with, but I am guessing that only one of them becomes fused with the AF6 protein?

In their response to point 3, the authors do not address the question of whether the MLL protein dimerizes in vivo to any greater extent when fused to AF6 than it does by itself. If the authors believe that dimerisation is the key issue, then I think that providing in vivo data to show the MLL-AF6 dimerises more than does intact MLL is an important point.

Moving on to point 4, I think the authors will find, if they look back at previous structures, that other RAS effectors do, in fact, interact with and stabilise the Switch 2 region in Ras. For

example, they might like to look at the crystal structure of the PI3 kinase bound to RAS (Pakold, et al., Cell, 2000, 103, pages 931-4).

Finally, regarding point 7, the immuno-precipitation experiment that they refer to in Figure 6b, does not demonstrate that the MLL-AF6 and DOT1L proteins “physically interact” with each other. These sorts of experiments can be extremely confusing because immuno-precipitation of one protein often brings down substantial chunks of chromatin with it and then, when one probes for the second protein with another antibody, one can easily detect the second protein when it binds to a different nucleosome or other regions of chromatin nearby. These experiments unequivocally do not demonstrate a “physical” interaction between the two proteins. If one did a careful digestion of a chromatin to obtain a mono-nucleosome, you might be able to show that the two proteins bind to the same nucleosome but, even then, they do not necessarily show that there is a physical interaction between the two proteins.

In summary, I still think there is a lot of good work reported in this paper but, for someone like myself who is outside the field, it is very difficult to assess whether the conclusions that are drawn and the mechanisms they propose are really supported by the data and previous work in the field. The paper is simply not written or argued tightly enough. I would like to see the paper written much more clearly so that one can understand the biochemistry of MLL and their interpretation of the results.

Reviewer #3 (Remarks to the Author):

Evolution of AF6-RAS Association and its Implications in Mixed-Lineage Leukemia by Matthew J. Smith et al. describes a different mode of action between the dimer of the fusion protein MLL-Arf6 and the monomeric forms of the independent components. Besides structural and biochemical characterization they perform an affinity MS approach to identify interacting partners of the different protein forms, which I was asked to review.

The Bio-ID approach is chosen here because of the difficulty of purifying the MLL protein endogenously from the nucleus under mild conditions, maintaining the interaction partners. The approach overall seems to be performed well revealing 199 “high confident interactors” of which 2 could be validated by immunoprecipitation and western blot.

Some points that need clarification:

The authors state in their rebuttal to have been “extremely stringent” in selecting the interaction partners. This is somewhat overstated as there were only 2 bio-replicates and the used FDRs are standard in interactomics studies. More importantly, the description of the actual experiment is vague. In the text it is stated there were 2 bio-replicates for the actual samples of interest (this info is absent in the methods section) but it’s very difficult to find how exactly the control experiments are performed. The authors describe 8 ‘control runs’ but when reading carefully there are 2 controls (cells expressing the BirA*/Flag tag only, and BioID analysis conducted on an unrelated bait protein (GFP)), which are both injected 4 times, resulting in 8 runs? So these are actually single control experiment and there are no bio-replicates of the control? This should be clarified in the methods section.

In Figure 6 it’s unclear what the different colors mean for the protein nodes; yellow, green, orange, reddish? Also the size of the nodes seems to differ? Is there a reason for this?

The construction of the actual network is not well described. It is stated in the figure legend that: “Partners specific for dimer or monomer MLL proteins were enriched with data from iRefIndex (specifically IntAct, BioGRID, MINT, and DIP) and imported to Cytoscape.” What does that mean for the network? How many proteins are added and how many of the original 199 interactors are in the network. The suppl excels are missing. This should be described better.

The number of validated proteins on the total number of interactors seems low, even though the immunoprecipitation is hard to perform, as stated by the authors, they got it to work so why were not any other proteins validated or at least the ones stated as “key hub proteins”?

Reviewer #4 (Remarks to the Author):

MLL fusion proteins generally cause poor prognosis leukemias in humans. MLL-AF6 stands out among MLL leukemias for two reasons: 1) Along with AF10, MLL-AF6 adult AML has the worst prognosis of all MLL fusion leukemias (Grimwade et al, Blood 2010; Balogobind et al, Blood 2009) and 2) although considered to be one of the “common” fusion partners (Meyer et al, Leukemia, 2013), unlike AF10, AF4, ENL, AF9 or ELL, AF6 is not part of a transcription elongation complex and thus its function is generally unknown. For these two reasons, understanding more about the function of MLL-AF6 is both important and interesting.

Here, the authors have shown that RA1, a canonical RAS binding domain in AF6, is partially dependent on an N terminal alpha-helical domain for tight association with RAS. MLL-AF6 fusions truncate this N terminal domain, exposing hydrophobic residues in RA1 and turning it into a self-association domain. This dimerization causes MLL-AF6 to become associated with the DOT1L H3K79 methyltransferase. Overall, this is a well performed and very interesting study and it proposes a novel mechanism for how a truncated AF6 protein can induce dimerization of MLLN.

I think for the most part the authors have answered the issues raised by the previous reviewers, but I think that there are still a few outstanding issues as outlined below:

Major points

1. In its current form, the paper still does not give proper due to past work on the dimerization of MLL in leukemia. For example, it is essential that they reference work from the Rabbits lab (Dobson et al, EMBOJ, 2000) that originally proposed that simple oligomerization of MLLN was sufficient to induce leukemogenesis. The Dobson paper is mentioned in the response to reviewers but for some reason I can't find this reference in the main paper. It really should be in the introduction as the first paper to propose the dimerization model for MLL fusions. It should also be mentioned in the introduction that two immediate follow up studies set out to directly test

this idea and were able to show that dimerization is sufficient for immortalization (Martin et al, Can Cell, 2003; So et al Can Cell 2003), providing a useful proof of principle for this concept.

2. MLL-AF6 has been shown to be strongly associated with H3K79 methylation by ChIP-seq, and the authors here show that dimerization induced by AF6 causes MLL-AF6 to be associated with DOT1L. This is a very interesting and novel result and provides a good explanation for the ChIP-seq data. However, this result should be discussed within the context of work that suggests dimerization of MLLN alone is not sufficient for increased H3K79me2 levels at gene targets (Milne et al, Cancer Res. 2005;65(24):11367-74). Using an inducible FKBP dimerization domain, researchers showed that although simple dimerization of MLLN was sufficient to transform cells (Martin et al, Cancer cell 2003) and dimerization caused MLL-FKBP binding to downstream target genes, this was not sufficient for increased H3K79me2 levels and that transformation occurred via some other mechanism, potentially acetylation of histones (Milne et al, Cancer Res. 2005;65(24):11367-74). This result may have something to do with the relative strength of dimerization induced by the FKBP domain, but the work from Milne et al suggests that dimerization does not automatically create an association with DOT1L and this should be discussed in the context of the MLL-AF6 results here.

3. The authors state “The final common genetic anomaly driving MLL leukemia’s is a partial tandem gene duplication (PTD) that produces a fusion of two MLL N-termini, effectively replicating dimerization”. This is an interesting potential point (that has been suggested by others previously), but is there any evidence that MLL-PTD leukemias have similar gene expression profiles as MLL-AF6 or other MLL fusion protein leukemias? My understanding is that MLL-PTD gene expression profiles are distinct from other MLL leukemias, suggesting that there is an alternate underlying mechanism.

4. Martin et al (2003) and So et al (2003) along with many others have already proposed that MLL fusion partners can be divided into two groups: nuclear and cytoplasmic, and that cytoplasmic partners are more likely to contain dimerization domains. In the discussion, the authors state “Here we provide a molecular mechanism of MLL activation resulting from its fusion with the RAS effector AF6, and evidence that oligomerization is a near universal property of MLL partners normally localized outside the nucleus.” It is great that the authors robustly tested this, but credit should also be given to past work that has already suggested that this would be true.

Minor points

1. I would be interested to hear speculation from the authors on how MLL-AF6 may interact with DOT1L without also interacting with other components of the DOT1L complex such as ENL (this lack of interaction between MLL-AF6 and ENL was shown in Yokoyama et al, *Cancer Cell* 17, 2010).
2. The authors should mention that “dimerizing” fusions, not including MLL-AF6, only represent a small number (about 10% or so?) of MLLr patients, so it is not entirely clear how “generalizable” these results with AF6 really are. I think the novelty of this work is more about MLL-AF6 itself, which is the most aggressive clinically of all the MLL-FPs, and the fact that dimerization promotes association with DOT1L.
3. “Isothermal titration calorimetry (ITC) was used to measure an affinity of AF6 RA1 for RAS-GMMPNP (Fig. 2b).” should refer to supplementary Figure 2b.

Rebuttal

Reviewer #1 (Remarks to the Author):

The authors for the most part have addressed the comments raised in prior review. However, there is one point that has not been appropriately addressed. In the introduction, the authors state “It has been proposed that AF6-mediated MLL activation is dependent on its first RAS association (RA) domain (ref 15), and ...” The referenced study actually demonstrated that the RA1 domain was the minimal region sufficient for oncogenesis in structure/function experiments as encapsulated in the title of the publication “Self-association mediated by the Ras association 1 domain of AF6 activates the oncogenic potential of MLL-AF6”. Thus, the sentence needs to be revised to state “It has been demonstrated that AF6-mediated MLL activation is dependent on its first RAS association (RA) domain.....”.

This has been corrected in the text.

Reviewer #2 (Remarks to the Author):

- 1. Thank you for sending me the revised version of the manuscript entitled “Evolution of AF6-RAS Association and its Implications in Mixed-Lineage Leukemia” Sadly, the authors have not addressed the questions raised from the first version of their manuscript, and I am still not convinced by the molecular mechanism that the authors are proposing for MLL activation resulting from its fusion with AF6. They believe that they “provide evidence that oligomerisation is the dominant mechanism driving oncogenesis from rare MLL translocation partners”. Their mechanism may be correct, but whilst I think they do provide evidence that deletion of the alpha-N-helix in the Ras-binding of AF6 could lead to dimerisation of MLL-AF6, it doesn't appear to be a particularly tight interaction. It is therefore not clear whether MLL-AF6 exists predominantly as a dimer in vivo, and I do not think the results necessarily imply that it is dimerisation that leads to the recruitment of DOT1L etc. Deletion of the alpha-N-helix clearly results in the exposure of a sticky hydrophobic patch on the surface of the MLL-AF6 protein. This could easily result in the aberrant recruitment of other proteins or complexes, depending on the relative amounts of monomer and dimer in vivo. Notably, even at high concentrations in vitro most of the AF6 RA1 domain protein is monomeric – see Figure 1b. Why do the authors think it is dimerisation of MLL-AF6 that leads to the recruitment of DOT1L etc.? I still think an equally plausible mechanism is that the N terminal domain of the MLL proteins in some way targets the protein to chromatin and its fusion with AF6 results in the aberrant recruitment of proteins such as DOT1L to MLL sites via this sticky patch, and that is what leads to transformation.*

We thank the reviewer for comments on our proposed molecular mechanism, and an offer to further clarify our rationalization. The key concern is that exposure of a hydrophobic patch that nucleates dimerization *in vitro* may not necessarily function by the same mechanism *in vivo*. Instead, the reviewer proposes that a ‘sticky patch’ composed of exposed hydrophobic residues could recruit DOT1L/ELL/etc. to chromatin sites following targeting by the MLL N-terminus. We would make the following points to support our model:

- As discussed in our manuscript and extensively in the previous rebuttal (to reviewer 1), the first evidence that dimerization can activate MLL came serendipitously, *via* observation that *Mll-lacZ* caused leukemia in mice¹. Bacterial lacZ/ β -galactosidase is a constitutive tetramer. Subsequent work demonstrated that inducible, synthetic FKBP dimers can cause leukemia², and the fusion partners GAS7^{3,4}, AF1p/EPS15⁴, SEPT6⁵, EEN/SH3GL1⁶ and GPHN⁷ have predicted dimerization motifs which can activate MLL activity. Thus, it is recognized that oligomerization by several mechanisms distinct from the hydrophobic patch can activate MLL.
- Our bioinformatic analysis of 36 cytosolic partners of MLL identified oligomeric potential in 34 of these that would also not depend on exposure of hydrophobic residues. This is consistent with the preceding point: all previously studied MLL oligomerizing fusions function by coiled-coils (or synthetic oligomers in the case of lacZ¹ or FKBP²).
- Wild-type MLL function depends on a set of PHD fingers in the central region of the protein which are known to oligomerize⁸. The PHD domains are absent from MLL fusions as the breakpoint cluster region lies immediately upstream.
- Transcribed MLL fusion proteins are localized in densely-packed nucleolar regions, and the presence of high MLL-AF6 concentrations with complementary exposed patches and a potential to dimerize, even if only at μ M affinity, means the probability of *in vivo* dimerization would be high.
- A total of 7 BioID hits were identified only using MLL-AF6^{RA1} (dimer) that were not present in MLLN alone or MLL-AF6 ^{α N/RA1} (monomers). These were ELL, DOT1L, ICE1, AF17, AF10, URB1 and PHGDH. Several of these are known to be important for MLL-mediated H2K79 methylation (DOT1L, AF17, AF10) or transcriptional elongation (ELL). Further, MLL-AF6 is susceptible to inhibition of DOT1L activity⁹. The likelihood that a ‘sticky patch’ would non-specifically recruit the very proteins required for MLL-induced gene activation is small. Instead, this must be an intrinsic characteristic of the fusion proteins themselves and all evidence supports a significance for oligomerization.
- Data we were asked to remove from the first submission concerned the question of why *MLL* breakpoints with *AF6* are distinct from those of other translocations. Specifically, *AF6* is almost always fused to *MLL* intron 9 while fusion of *AF9*, *AF4*, *AF10*, *ELL* and *ENL* distributes evenly across introns 9, 10 and 11 (intron 10 is smaller):

The C-terminal ends (last 30 residues) encoded by exons 10 and 11 are highly concentrated in hydrophobic residues (12/30 and 16/30 respectively), while the end of exon 9 encodes only 5/30 hydrophobic residues:

As these residues would be fused immediately adjacent to the AF6 RA1 domain exposed hydrophobics, we postulated that the hydrophobic nature of exon 10 or exon 11 would inhibit dimerization (akin to the α N helix in wild-type AF6). Indeed, MLL/exon 9-AF6 fusion is a dimer while the MLL/exon 10-AF6 fusion is a monomer:

The suggestion that a ‘sticky patch’ is recruiting DOT1L to MLL, rather than dimerization, is not supported by these results. Fusion of AF6 to MLL exon 10 or exon 11 would actually be selected if this were the case, as the ‘sticky patch’ would be much larger. Conversely, we show that these translocations would encode a fusion protein with reduced capacity for dimerization and that correlates well with the atypically frequent exon 9-AF6 fusions, which support dimerization.

Taken together, there is overwhelming support for the molecular mechanism we’ve proposed – oligomerization – and little to suggest exposure of a hydrophobic patch could specifically nucleate complex of MLL partner proteins required for transcription/epigenetic marks.

2. *I do not think the authors provide any clear evidence that “oligomerisation” is the dominant mechanism. The authors do not address the question of whether the MLL protein dimerizes in vivo to any greater extent when fused to AF6 than it does by itself. If the authors believe that dimerisation is the key issue, then I think that providing in vivo data to show the MLL-AF6 dimerises more than does intact MLL is an important point.*

Unfortunately, there are no available approaches whereby we could study *in vivo* dimerization of the 340 kDa MLL-AF6 protein in the nucleolus from a biophysics standpoint. We would welcome any suggestions on how this could be accomplished experimentally. The wild-type MLL protein is 432 kDa. Simply cloning the cDNA’s encoding the fusion proteins used for these studies is not trivial. In lysates, the 340 kDa MLL-AF6 protein is difficult to transfer even from 6% SDS-PAGE gels. In fact, this is what makes our molecular understanding of the RA1 dimer interface so important – it allowed the first description of an MLL dimer vs a monomer by

comparisons with MLL-AF6 in which we re-inserted the α N helix (to prevent RA domain dimerization). The mouse studies and BioID proteomic studies we present are the first to compare monomer and dimer *in vivo*. In regards to assessing the *in vivo* potential for MLL fusion proteins to dimerize, the only options would be indirect measurements by FRET/BRET/PCA or by co-precipitation of differentially tagged proteins. We can observe no visible differences in the localization of Venus-tagged MLLN, MLL-AF6, MLL-AF6^{RA1} or MLL-AF6 ^{α N/RA1}.

Due to this, we do not deem FRET/BRET/PCA-type assays would be informative as these proteins are clearly localized to the densely packed nucleolar region (speckles) and all would give positive co-localization signals. To satisfy the reviewer's request, we therefore attempted to determine whether these proteins could self-associate using co-precipitations of differentially tagged proteins. Indeed, the only existing evidence, to our knowledge, that any MLL proteins (wild-type or leukemogenic fusion) are able to self-associate after nearly two decades of intense study comes from the following three figures^{4,5,10}:

At right, note that co-precipitation of Flag-MLL-AF6(RA1) and HA-MLL-AF6(RA1) from Liedtke *et al.*¹⁰ does not control for background – this experiment should have been an anti-HA IP and anti-Flag immunoblot – thus we cannot draw conclusions from this figure. Having performed many immunoprecipitations now with large MLL-tagged nucleolar proteins we appreciate how difficult these experiment are, and Western blots of the quality presented in the above figures is typical. Nevertheless, to answer the reviewers question we tagged MLL-AF6 (full length), MLL-AF6^{RA1}, MLL-AF6 ^{α N/RA1} and the MLL N-terminal half alone (MLLN) with

either GFP or Flag and co-transfected the differentially tagged proteins. Cells were lysed in a modified RIPA buffer (50 mM Tris (pH 7.5), 1% NP-40, 1% sodium deoxycholate, 10% glycerol, 300 mM NaCl, 2 mM EDTA, 1 mM DTT and protease inhibitors) supplemented with benzonase nuclease to remove all nucleic acids (incubation for 1 hour). Cleared lysates were then immunoprecipitated with an anti-GFP antibody and these were probed for the presence of Flag-tagged protein by Western blot. GFP alone served as a control. Our results are presented in the new Fig. 7b and at the end of the Results section *Cytoplasmic MLL Fusion Partners*. We observe that full length MLL-AF6 and also the RA1 and α N-RA1 variants are all co-precipitated from cell lysates. Importantly, we consistently detect that the MLLN fragment alone is effectively precipitated from lysates by differentially tagged MLLN. All IPs were performed in the alternative direction (i.e. anti-Flag IP followed by anti-GFP immunoblot) with the identical result. As we are confident that benzonase treatment effectively prevents nucleic acid cross-linking, we considered alternative explanations as to why MLLN alone appears to ‘self-associate’ in cell lysates. Interestingly, the structure of the most well-characterized, constitutive binding partners of the MLL N-terminus – MEN1 and LEDGF¹¹⁻¹⁵ – with an MLL fragment are derived from a co-crystal in which these proteins form a dimeric complex¹⁶:

Further, Vanderlinden *et al.* have convincingly shown that the LEDGF protein itself also exists as a dimer, likely through a domain-swapped dimerization of its PWWP domain (not present in the above structure)¹⁷. As we demonstrate in Fig. 6b, MLLN is sufficient for MEN1 association. Thus, we would propose that cellular proteins or complexes such as MEN1/LEDGF can cross-link the N-terminal half of MLL in the absence of any fusion. Additionally, other proteins identified as MLLN interactors in our BioID experiment (Fig. 5b) may have similar capacities. This demonstrates the difficulty in drawing biophysical conclusions from *in vivo*-derived biochemical data, and again why we do not believe that AF6 RA1 dimerization had been effectively demonstrated previously. Comprehensive *in vitro* experiments utilizing MLLN and its fusion with various dimerizing partners, if feasible, will be required in the future to shed light on the biophysical consequences of fusion-mediated dimerization of MLLN. The dimer partners could modulate MLLN activity by inducing a conformational change, or simply *via* increased affinity/avidity of dimer MLLN to other cellular partners.

3. *In their rebuttal the authors propose “that an MLL fusion partner such as AF6 can mediate homo-dimerization, analogous to the PHD fingers, but the regulatory activity imparted by the wild-type MLL PHD domains is lost and the epigenetic activity becomes constitutive”. This also does not make sense to me. Why do they think the epigenetic activity becomes constitutive when the enzymatic domain that mediates H3K4 methylation (the SET domain) is also lost in this fusion? What epigenetic activity are we talking about? It is possible that by reading the literature in this area in detail I could perhaps understand what is going on here, but I had hoped that the authors would come back (at least in the rebuttal) with a more detailed analysis of their data in the context of the existing literature and provide a solid argument of why they think their mechanism is preferred. AND*
4. *I do not want to get into an argument about the pros and cons of different proteomics methods, but I do think it important that they point out that there is a family of MLL complexes containing different MLL proteins. Maybe all the literature is wrong, but there seem to be many papers describing WDR5 as a component of different MLL complexes (see e.g. Hubert et al., Epigenetics. 2013, 8, 79-91). It is also not clear in the text/figures which MLL protein they are actually working with, but I am guessing that only one of them becomes fused with the AF6 protein?*

We will attempt to answer the concerns in points 3 and 4 together, as they are related and require some background on *MLL* translocation-induced leukemias. First, the mammalian *MLL* family of histone lysine methyltransferases contains 6 members (*MLL1–4*, *SET1A* and *SET1B*)¹⁸. Only *MLL1* (also *MLL* or *KMT2A*) is involved in chromosomal translocations that induce leukemias, and this has been an enormous field of study since its discovery in the late 90s. All wild-type *MLL* family proteins have associated H3K4 methyltransferase activity mediated by the C-terminal SET domain – except *MLL1*, which demonstrates only low level activity¹⁹. To augment this, *MLL1* (and also other family members) associate with *WDR5*, *RBBP5* and *ASH2L* in a biochemical complex mediated directly by the SET domain¹⁸. Despite this, as you can see from Figure 1a in our manuscript, the complete C-terminal half of *MLL* is lost in all translocations. It has been clearly demonstrated that this region is dispensable for *MLL*-driven leukemogenesis²⁰. Thus, while it is true that *WDR5* is a component of different *MLL* complexes *this is not true of leukemogenic MLL fusion proteins* as the SET domain is no longer present. The ‘constitutive epigenetic activity’ of *MLL* fusions does not come from the wild-type *MLL* SET domain or association with *WDR5* (as both are lost), but from induced partner proteins like the *DOT1L* methyltransferase^{21–23}. As such, epigenetic marks switch from H3K4 to the *DOT1L*-specific H2K79. With regards to the PHD domains – these are readers of H3K4 methylation and would normally function in targeting *MLL* to sites containing these marks^{13,14}. Additionally, they are known to oligomerize, which has been suggested to play a role in *MLL*-mediated gene expression⁸. As with the SET domain, the PHD domains are lost in all *MLL* fusions (see Figure 1a for *MLL*-AF6). This implies that while partner proteins fused to *MLL* may comprise a new oligomerization site, replicating function of the lost PHD domains, they would lose the normal targeting function of the PHDs and would no longer be sequestered at H3K4 marked sites. We hope that this clarifies some existing information for the reviewer, and puts our data in better context. It is difficult to account for the vast literature on this subject in a manuscript of limited size, but we have attempted to convey the above points in our resubmission (Introduction, paragraph 1; Discussion: Interactors of AF6-Induced *MLL* Dimers, paragraph 1).

5. *I think the authors will find, if they look back at previous structures, that other RAS effectors do, in fact, interact with and stabilise the Switch 2 region in Ras. For example, they might like to look at the crystal structure of the PI3 kinase bound to RAS (Pakold, et al., Cell, 2000, 103, pages 931-4).*

The reviewer is correct and PI3K would be the third example of a RAS binding domain that contacts the RAS switch II region. This is mediated by a residue (K234) in the β 2 strand rather than an N-terminal α -helix and is a unique feature of the PI3K interaction not related to the α N helix identified here. We have noted this in the text, Results: Detection of an α N Helix, paragraph 2.

6. *Finally the immuno-precipitation experiment that they refer to in Figure 6b, does not demonstrate that the MLL-AF6 and DOT1L proteins “physically interact” with each other. These sorts of experiments can be extremely confusing because immuno-precipitation of one protein often brings down substantial chunks of chromatin with it and then, when one probes for the second protein with another antibody, one can easily detect the second protein when it binds to a different nucleosome or other regions of chromatin nearby. These experiments unequivocally do not demonstrate a “physical” interaction between the two proteins. If one did a careful digestion of a chromatin to obtain a mono-nucleosome, you might be able to show that the two proteins bind to the same nucleosome but, even then, they do not necessarily show that there is a physical interaction between the two proteins.*

As detailed in our Methods section, transfected cells were lysed in a modified RIPA buffer (50 mM Tris (pH 7.5), 1% NP-40, 1% sodium deoxycholate, 10% glycerol, 300 mM NaCl, 2 mM EDTA, 1 mM DTT, 5 mM MgCl₂, nuclease and protease inhibitors). The high salt concentration dissociates proteins from chromatin and the presence of nuclease degrades all nucleic acids. Specifically, we use the benzonase nuclease from Sigma (product number E1014), which is: “a genetically engineered endonuclease from *Serratia marcescens* that degrades all forms of DNA and RNA (single stranded, double stranded, linear and circular) and is effective over a wide range of operating conditions...ideal for removal of nucleic acids from recombinant proteins and for applications where complete digestion of nucleic acids is desirable.” To the reviewers point, we do not believe chromatin to be present in our extracts on which we perform co-IPs, and we would argue our results do demonstrate a biochemical interaction. Combined, we have extensive experience in performing co-immunoprecipitations with a variety of exogenous or endogenous proteins (nuclear/cytoplasmic, large/small, tagged/untagged, etc.) under a wide variety of conditions and believe our results to be accurate.

Reviewer #3 (Remarks to the Author):

1. *Evolution of AF6-RAS Association and its Implications in Mixed-Lineage Leukemia by Matthew J. Smith et al. describes a different mode of action between the dimer of the fusion protein MLL-AF6 and the monomeric forms of the independent components. Besides structural and biochemical characterization they perform an affinity MS approach to identify interacting partners of the different protein forms, which I was asked to review. The Bio-ID approach is chosen here because of the difficulty of purifying the MLL protein endogenously from the nucleus under mild conditions, maintaining the interaction partners. The approach*

overall seems to be performed well revealing 199 “high confident interactors” of which 2 could be validated by immunoprecipitation and western blot. Some points that need clarification: The authors state in their rebuttal to have been “extremely stringent” in selecting the interaction partners. This is somewhat overstated as there were only 2 bio-replicates and the used FDRs are standard in interactomics studies. More importantly, the description of the actual experiment is vague. In the text it is stated there were 2 bio-replicates for the actual samples of interest (this info is absent in the methods section) but it’s very difficult to find how exactly the control experiments are performed. The authors describe 8 ‘control runs’ but when reading carefully there are 2 controls (cells expressing the BirA/Flag tag only, and BioID analysis conducted on an unrelated bait protein (GFP)), which are both injected 4 times, resulting in 8 runs? So these are actually single control experiment and there are no bio-replicates of the control? This should be clarified in the methods section.*

We thank the reviewer for these comments on our proteomic work and will clarify on the points raised. As indicated by the reviewer, the FDRs used in our data analysis follow standards in the field. With regards to the replicates, each of the two biological replicate purifications is analyzed with SAINT individually and we are using the Averaged SAINT score (across both biological replicates) for the establishment of the final list and the estimation of the false discovery rates. This ensures that the interactions we are reporting have been detected across both of the replicates. We should have been clearer on the description of the negative controls. The reviewer is correct in that we have used two different types of negative controls. The first type of control (BirA*-FLAG only) is meant to mimic the condition in which the expressed biotin ligase, BirA*, biotinylates non-specifically intracellular proteins. The second control (GFP) is meant to mimic the condition in which endogenous biotinylation (which primarily occurs on mitochondrial carboxyases) would be predominant. We should have mentioned in the methods (and have now done so) that – as with our true samples – each of the negative controls was analyzed in biological replicates, in this case with four independent biological replicates per type of control. (These are not simple re-injections or technical replicates). In our experience, performing these different biological replicates is important, as the generation of cell pellets and the affinity purification can indeed vary from batch to batch and we are trying to make sure we capture this biological variability for both the bait and the control purifications. We also should have been clearer in the definition of the control compression using SAINTexpress: as was previously shown in Mellacheruvu *et al.* (Nature Methods, 2013)²⁶, a compression strategy in which the highest x counts across all controls selected is used for calculating the interaction significance helps to capture spurious binding behavior of some contaminants. In our case, because we are using 2 different types of controls we elected to compress to the minimum number of controls in any one group, which was 4. This means that each potential prey across the two biological replicates of the bait is assessed for significance across the 4 highest values across the 8 controls we used here. We have re-written this section in the Methods more clearly and also point the reader to the manuscript by Lambert *et al.* (J Proteomics, 2015)²⁷ to provide additional benchmarking for this approach.

2. *In Figure 6 it’s unclear what the different colors mean for the protein nodes; yellow, green, orange, reddish? Also the size of the nodes seems to differ? Is there a reason for this?*

The reviewer is correct and we wish to be clearer in how we treated our data. The MLL monomer/dimer network, generated as described below (point 3), was analyzed with Cytoscape to generate Figure 6a. Specifically, protein interaction pairs were entered as an undirected network and interpreted with NetworkAnalyzer. There was no manual treatment of the data, the two subnetworks clustered inherently. Both the colour (red-yellow-green gradient) and size (large to small) of nodes were mapped using Betweenness Centrality (a measure based on shortest paths that determines the connectedness of a given node). Therefore, nodes that are highly connected in this network are coloured red and are larger, those that are not highly connected are coloured green and are smaller. Yellow are intermediate. GO CC descriptions for the most highly connected nodes are provided below the network map. For edges, their transparency was generated by mapping to Edge Betweenness (again based on shortest paths and a measure of connectedness). Thus, opaque edges are highly connected and more transparent edges are less connected to help visualize the core components of the network. Both edges and nodes were also bundled (number of handles=3, spring constant=0.003, compatibility threshold=0.3, maximum iterations=500). This information has been added to the figure legend to clarify how the network was analyzed, and we added a small legend.

- 3. The construction of the actual network is not well described. It is stated in the figure legend that: "Partners specific for dimer or monomer MLL proteins were enriched with data from iRefIndex (specifically IntAct, BioGRID, MINT, and DIP) and imported to Cytoscape." What does that mean for the network? How many proteins are added and how many of the original 199 interactors are in the network. The suppl excels are missing. This should be described better.*

This again refers to the network in Figure 6a describing monomer vs. dimer-specific interactors. To construct this network we selected all BioID hits specific for MLLN alone and the MLL-AF6 ^{α N-RA1} fusion. These represent preys that associated only with monomeric MLL. We also selected those found only in the MLL-AF6^{RA1} dataset, being specific for dimeric MLL. These (13) monomer- and (7) dimer-specific interactors are from Figure 5b and 5c. Known interactions with these 20 hits from the iRefIndex database (IntAct, BioGRID, MINT, and DIP) were used to extend the network for each of the monomer and dimer-specific interactors, presented in Supplemental Tables 2 and 3. These interactions were input to Cytoscape for mapping the network in Figure 6a as described above (point 3).

- 4. The number of validated proteins on the total number of interactors seems low, even though the immunoprecipitation is hard to perform, as stated by the authors, they got it to work so why were not any other proteins validated or at least the ones stated as "key hub proteins"?*

To comply with the reviewer's request we have attempted to verify several of the relevant interactions identified by BioID. Our interest for this manuscript is hits related specifically to MLL-AF6^{RA1} dimers that were absent from MLLN alone/MLL-AF6 ^{α N-RA1} monomers (in Fig. 5c). In particular, we focused on the MLL leukemia relevant targets AF10, AF17 and ELL. The DOT1L and MENIN co-precipitations were performed with commercial antibodies against endogenous proteins (Fig. 6b). We attempted similar co-precipitations with AF10, AF17 and ELL using antibodies purchased from 2 sources (Santa Cruz or Abcam), but unfortunately found the antibody reagents to be of poor quality and these proteins at very low abundance. We thus

acquired cDNAs for these hits and cloned them into mammalian expression vectors with epitope tags (Flag or GFP). We attempted to co-express these with our complete set of MLL-AF6 fusions and controls but were not able to detect co-precipitation (at least, not at high intensity or consistency as with the DOT1L and MENIN interactions). Below are the resulting immunoblots:

These co-precipitations are done under very stringent conditions (1% NP-40, 1% sodium deoxycholate, 10% glycerol, 300 mM NaCl), a necessity to extract the proteins from the nucleolar region. It is therefore expected that only high affinity interactions would be preserved, our original rationale for using BioID, so we sought an alternative experimental approach to confirm these data. We considered using biotin and the BirA tag ‘proximity sensor’ to directly quantitate levels of biotinylation on these specific targets. We co-expressed BirA/Flag-MLL-AF6^{RA1} (dimer) or BirA/Flag-MLL-AF6^{aN-RA1} (monomer) with GFP-tagged ELL, AF17 or AF10 in cells growing in media supplemented with biotin (24 hr), and baits were immunoprecipitated from lysates with anti-GFP. Western blots were probed with streptavidin-HRP to directly detect biotinylation levels on the baits, and blots were then stripped and re-probed to visualize loading. As demonstrated in the new Fig. 6c and in the Results section *Interactors of AF6-Induced MLL Dimers*, the addition of the α N-helix between MLL and AF6 significantly and consistently resulted in lower levels of AF17, AF10 and ELL biotinylation, whereas MENIN was repeatedly biotinylated uniformly. Interestingly, we clearly observe a low intensity band in streptavidin blots that likely corresponds with MLL-AF6^{RA1} co-precipitation with GFP-AF17 and GFP-AF10 (not present in the MLL-AF6^{aN-RA1} lane or in ELL IPs), but could not confirm this with anti-Flag (the streptavidin probe being much more sensitive). Thus, we are confident these results further corroborate our BioID data and they nicely demonstrate that incorporation of the α N-helix (and inhibition of MLL-AF6 dimerization) reduces association with these relevant components of DOTCOM and AEP.

Reviewer #4 (Remarks to the Author):

MLL fusion proteins generally cause poor prognosis leukemias in humans. MLL-AF6 stands out among MLL leukemias for two reasons: 1) Along with AF10, MLL-AF6 adult AML has the worst prognosis of all MLL fusion leukemias (Grimwade et al, Blood 2010; Balogobind et al, Blood 2009) and 2) although considered to be one of the “common” fusion partners (Meyer et al, Leukemia, 2013), unlike AF10, AF4, ENL, AF9 or ELL, AF6 is not part of a transcription elongation complex and thus its function is generally unknown. For these two reasons, understanding more about the function of MLL-AF6 is both important and interesting. Here, the

authors have shown that RAI, a canonical RAS binding domain in AF6, is partially dependent on an N terminal alpha-helical domain for tight association with RAS. MLL-AF6 fusions truncate this N terminal domain, exposing hydrophobic residues in RAI and turning it into a self-association domain. This dimerization causes MLL-AF6 to become associated with the DOT1L H3K79 methyltransferase. Overall, this is a well performed and very interesting study and it proposes a novel mechanism for how a truncated AF6 protein can induce dimerization of MLLN. I think for the most part the authors have answered the issues raised by the previous reviewers, but I think that there are still a few outstanding issues as outlined below: Major points:

- 1. In its current form, the paper still does not give proper due to past work on the dimerization of MLL in leukemia. For example, it is essential that they reference work from the Rabbits lab (Dobson et al, EMBOJ, 2000) that originally proposed that simple oligomerization of MLLN was sufficient to induce leukemogenesis. The Dobson paper is mentioned in the response to reviewers but for some reason I can't find this reference in the main paper. It really should be in the introduction as the first paper to propose the dimerization model for MLL fusions. It should also be mentioned in the introduction that two immediate follow up studies set out to directly test this idea and were able to show that dimerization is sufficient for immortalization (Martin et al, Can Cell, 2003; So et al Can Cell 2003), providing a useful proof of principle for this concept.*

We thank the reviewer for these suggestions and have referenced the appropriate work in the introduction, paragraph 2.

- 2. MLL-AF6 has been shown to be strongly associated with H3K79 methylation by ChIP-seq, and the authors here show that dimerization induced by AF6 causes MLL-AF6 to be associated with DOT1L. This is a very interesting and novel result and provides a good explanation for the ChIP-seq data. However, this result should be discussed within the context of work that suggests dimerization of MLLN alone is not sufficient for increased H3K79me2 levels at gene targets (Milne et al, Cancer Res. 2005;65(24):11367-74). Using an inducible FKBP dimerization domain, researchers showed that although simple dimerization of MLLN was sufficient to transform cells (Martin et al, Cancer cell 2003) and dimerization caused MLL-FKBP binding to downstream target genes, this was not sufficient for increased H3K79me2 levels and that transformation occurred via some other mechanism, potentially acetylation of histones (Milne et al, Cancer Res. 2005;65(24):11367-74). This result may have something to with the relative strength of dimerization induced by the FKBP domain, but the work from Milne et al suggests that dimerization does not automatically create an association with DOT1L and this should be discussed in the context of the MLL-AF6 results here.*

The discrepancy here is that our data indicate that MLL-AF6 dimers associate with DOT1L to invoke H3K79 methylation, and this corroborates previous hypotheses, but that MLL-FKBP dimers do not elicit a similar histone methylation pattern. As such, the reviewer implies that not all dimer-driven MLL fusions associate with the DOT1L methyltransferase. This requires some context regarding the MLL-FKBP fusion – this is an inducible, recombinant MLL fusion protein based on the 12 kDa FKBP domain and a synthetic molecule (‘dimerizer’) that is supplemented continuously in media. These studies (Martin et al., 2003 and Milne et al., 2005^{2,4}) were vital

steps in proof of concept that MLL dimerization could immortalize hematopoietic cells. With regards to the fusion itself, some important points:

- The AP20187 dimerizer has an EC_{50} in the low nanomolar range (10-20 nM)²⁸ and was generally used at 50 nM (up to 500 nM). While no dose-response profile was provided, we can assume due to the high affinity of AP20187 for the FKBP domain that in the presence of dimerizer the MLL fusions are constitutive dimers, though this would be extremely difficult to demonstrate experimentally.
- Compared to MLL-AF9 cells, “*MLL-FKBP-transduced cells grew slowly and required both IL-3 and SCF factor for growth initially in liquid culture*” and “*MLL-FKBP gave rise to monoclonal integrations, implying a lower efficiency (transformation)*”⁴.
- The MLL-FKBP fusion was not shown to induce leukemia in mice and the transformation imposed by MLL-FKBP is actually reversible⁴.
- Expression of several key genes (*Hox*, *Meis1*) in MLL-FKBP transformed cells are dependent on the presence of dimerizer, but the cells can grow in its absence (i.e. become dimerizer-independent), implying secondary genetic hits.
- MLL-FKBP dimerization has minimal effects on K79 methylation at *Hoxa9* and a minor effect at *Meis1*, this is in contrast to MLL-AF6 which “*display markedly high levels of histone 3 at lysine 79 (H3K79) demethylation in murine MLL-AF6 leukemias*” (Deshpande *et al.*, Blood, 2013)⁹, consistent with the other MLL fusions examined²⁹.

Taken together, it is difficult to compare MLL fusion to a natural partner such as AF6 with the inducible FKBP system. AF6 would undergo monomer-dimer conversion, which may be important to DOT1L binding. Indeed, self-association of the intrinsic MLL PHD fingers appears to be of low affinity, though it has not been properly measured (Fair *et al.*, MCB 2001)⁸. Unfortunately, other alleged dimer-mediated MLL fusions (GAS7, AF1p/EPS15, SEPT6, EEN/SH3GL1 and GPHN) have not been studied in the context of DOT1L association or H2K79 methylation. To completely resolve the reviewers point will require a systematic analysis of MLL-fusion-induced H2K79 methylation, a comprehensive biochemical and biophysical characterization of how DOT1L associates with MLL fusion complexes (direct or indirect), and a consistent approach using ChIP and RNA-seq in human cells harbouring MLL fusions. While these data will take several years to resolve, we have conveyed the main point raised by the reviewer (lack of DOT1L-mediated H2K79 methylation in MLL-FKBP) in the Discussion, paragraph 1.

3. *The authors state “The final common genetic anomaly driving MLL leukemia’s is a partial tandem gene duplication (PTD) that produces a fusion of two MLL N-termini, effectively replicating dimerization”. This is an interesting potential point (that has been suggested by others previously), but is there any evidence that MLL-PTD leukemias have similar gene expression profiles as MLL-AF6 or other MLL fusion protein leukemias? My understanding is that MLL-PTD gene expression profiles are distinct from other MLL leukemias, suggesting that there is an alternate underlying mechanism.*

The *MLL-PTD* is distinctive as it encodes the only leukemogenic MLL protein that retains the C-terminus, comprising the SET domain, and presumably H3K4 methyltransferase activity. It was demonstrated by Dorrance *et al.* (J. Clin. Inv., 2006)³⁰ that a single *Mll-PTD* allele in mice increases histone H3 and H4 acetylation and histone H3K4 methylation in the *Hoxa7* and *Hoxa9* promoters, corresponding with an increase in their expression. In 2005, a global examination of

gene expression profiles from various leukemic cell lines demonstrated clustering of the Eol-1 line (*MLL-PTD*) and the ML-2 line (*MLL-AF6*) (Andersson *et al.*, Leukemia)³¹. Moreover, two *MLL-PTD* leukemia cell lines are sensitive to DOT1L inhibition and demonstrate high levels of H2K79 methylation (Kühn *et al.*, Haematologica, 2015)³². Contrary to this, work in 2004 using microarray gene expression profiles suggested that AMLs containing *MLL-PTDs* do not cluster with AMLs driven by *MLL* chimeric fusions (Ross *et al.*, Blood)³³. A more recent comparative analysis of primary AML specimens challenges this, demonstrating significant overlap in the gene expression profiles of *MLL* fusion and *MLL-PTD* AMLs and identifying a common transcriptional profile (Lavallée *et al.*, Nat. Gen., 2015)³⁴. Altogether, it seems that *MLL* fusion and *MLL-PTD* cells do present similar gene expression profiles and seem to be driven by related molecular mechanisms, but this requires much further study. The likelihood is that significant heterogeneity exists within these samples. As the *MLL-PTD* does not relate directly to our work, and as the reviewer points out a potential similarity with dimer *MLL* has been alluded to previously, we have removed this supposition from our manuscript.

4. *Martin et al (2003) and So et al (2003) along with many others have already proposed that MLL fusion partners can be divided into two groups: nuclear and cytoplasmic, and that cytoplasmic partners are more likely to contain dimerization domains. In the discussion, the authors state “Here we provide a molecular mechanism of MLL activation resulting from its fusion with the RAS effector AF6, and evidence that oligomerization is a near universal property of MLL partners normally localized outside the nucleus.” It is great that the authors robustly tested this, but credit should also be given to past work that has already suggested that this would be true.*

The appropriate references have been added to the Discussion, paragraph 1.

Minor points:

1. *I would be interested to hear speculation from the authors on how MLL-AF6 may interact with DOT1L without also interacting with other components of the DOT1L complex such as ENL (this lack of interaction between MLL-AF6 and ENL was shown in Yokoyama et al, Can Cell 17, 2010).*

In the primary data referred to here (Yokoyama *et al.*; Canc. Cell, 2010)³⁵, the authors demonstrate that immunoprecipitation of *MLL* fused to *AF6* does not co-precipitate *AF4*, cyclin *T1*, *ENL* or *CDK9*. These proteins are also not precipitated by wild-type *MLL*. Conversely, *MLL-AF5q31*, *MLL-ENL* or *MLL-AF4* complex with these components of what the authors call *AEP* (*AF4*, *ENL* and *P-TEFb* complex). *AEP* is involved in transcriptional elongation and *MLL*-driven transformation, and they further demonstrate that this complex *does not* contain *DOT1L*. Others have shown that *DOT1L* *does* precipitate with some components of the *AEP* complex (i.e. Mohan *et al.*; Genes Dev., 2010)³⁶, and unfortunately there are incompatible reports concerning the exact biochemical components of both *AEP* and *DOTCOM* (*DOT1L* methyltransferase complex). An extensive domain mapping by Yokoyama *et al.* provides a molecular basis for the absence of *DOT1L* in *AEP* – both *AF5* and *DOT1L* use the same binding interface of *ENL* and are therefore incapable of associating simultaneously. Their suggestion is that *ENL* is involved in multiple subcomplexes, but could associate both with *AEP* and intermittently with *DOTCOM*. Thus, our detection of *MLL-AF6* complex with *DOT1L* is

consistent with Yokoyama *et al.*, as they found ENL and DOT1L to be components of distinct complexes. The second result of interest from their work is that MLL-AF6 and wild-type MLL require the AEP complex to promote transcriptional activity, in the absence of direct biochemical association, and co-localize with components of AEP at target sites on chromatin. We would agree with a two-step mechanism, as speculated by Yokoyama *et al.*, which separates activity of AEP and DOTCOM but promotes both H2K79 epigenetic marks and increased transcriptional activity. If dimerized MLL associates with DOTCOM and not directly with AEP, our data would suggest a slightly different model than theirs in that DOT1L methyltransferase activity would precede recruitment of RNAPII and AEP components to marked target sites. Further, in our response to Reviewer 3, point 4 we provide additional validation of MLL-AF6 association with other DOTCOM components.

2. *The authors should mention that “dimerizing” fusions, not including MLL-AF6, only represent a small number (about 10% or so?) of MLLr patients, so it is not entirely clear how “generalizable” these results with AF6 really are. I think the novelty of this work is more about MLL-AF6 itself, which is the most aggressive clinically of all the MLL-FPs, and the fact that dimerization promotes association with DOT1L.*

It is difficult to estimate exactly how many MLL fusions are driven by dimerization (even a partner like AF10 may function in this manner – Linder *et al.* J. Mol. Biol., 2000)³⁷, but we agree with the reviewer and will attempt to classify the known translocations. Frequency of the most common fusion partners in 1557 acute leukemia patients (Marschalek; Ann. Lab. Med., 2016)³⁸:

	AML	ALL	Total
AFF1/AF4	1%	60%	38.1%
MLLT3/AF9	29%	12%	18.3%
MLLT1/ENL	4%	18%	12.8%
MLLT10/AF10	15%	3%	7.1%
MLLT4/AF6	10%	1%	4.3%
ELL	11%	-	4.1%
PTD	11%	-	4.1%
Others	19%	6%	10.8%

The known ‘transcriptional elongators’ AF4, AF9, ENL, ELL and AF10 make up ~80% of cases, while the dimerizer MLL-AF6 makes up another 4-5%. The PTD, as discussed above in point 3, makes up another 4% of cases and could function in a similar manner as the dimerizers. The 10-11% of cases made of rare translocations can be broken down into ~36% nuclear and 64% cytoplasmic in origin (of the wild-type protein based on GO CC annotations, as in our Results section and calculated from the translocation partner database in Meyer *et al.*; Leukemia, 2013³⁹). We will roughly extrapolate that most of the cytoplasmic partners function by dimerization, as we demonstrate in this manuscript, and discard the nuclear proteins (though it is highly likely some of these function as dimerizers). This estimates that ~7% of the ‘Others’ from the chart above would function as dimer-inducers. With AF6, the total is 11% (with PTDs we would be considering 15% of MLL leukemias). We would argue this is a significant number of patients, made more so by the poor prognosis. Nevertheless, the frequency is important and we have included the 11% estimate in the discussion, paragraph 1.

3. “Isothermal titration calorimetry (ITC) was used to measure an affinity of AF6 RAI for RAS-GMMPNP (Fig. 2b).” should refer to supplementary Figure 2b.

This has been corrected.

References

1. Dobson, C. L., Warren, A. J., Pannell, R., Forster, A. & Rabbitts, T. H. Tumorigenesis in mice with a fusion of the leukaemia oncogene Mll and the bacterial lacZ gene. *EMBO J* **19**, 843–851 (2000).
2. Martin, M. E. *et al.* Dimerization of MLL fusion proteins immortalizes hematopoietic cells. *Cancer Cell* **4**, 197–207 (2003).
3. So, C. W. *et al.* MLL-GAS7 transforms multipotent hematopoietic progenitors and induces mixed lineage leukemias in mice. *Cancer Cell* **3**, 161–171 (2003).
4. So, C. W., Lin, M., Ayton, P. M., Chen, E. H. & Cleary, M. L. Dimerization contributes to oncogenic activation of MLL chimeras in acute leukemias. *Cancer Cell* **4**, 99–110 (2003).
5. Ono, R. *et al.* Dimerization of MLL fusion proteins and FLT3 activation synergize to induce multiple-lineage leukemogenesis. *J. Clin. Invest.* **115**, 919–929 (2005).
6. Liu, H. *et al.* Functional contribution of EEN to leukemogenic transformation by MLL-EEN fusion protein. *Oncogene* **23**, 3385–94 (2004).
7. Eguchi, M., Eguchi-Ishimae, M. & Greaves, M. The small oligomerization domain of gephyrin converts MLL to an oncogene. *Blood* **103**, 3876–82 (2004).
8. Fair, K. *et al.* Protein Interactions of the MLL PHD Fingers Modulate MLL Target Gene Regulation in Human Cells. *Mol. Cell. Biol.* **21**, (2001).
9. Deshpande, A. J. *et al.* Leukemic transformation by the MLL-AF6 fusion oncogene requires the H3K79 methyltransferase Dot1l. *Blood* **121**, 2533–2541 (2013).
10. Liedtke, M., Ayton, P. M., Somervaille, T. C. P., Smith, K. S. & Cleary, M. L. Self-association mediated by the Ras association 1 domain of AF6 activates the oncogenic potential of MLL-AF6. *Blood* **116**, 63–70 (2010).
11. Yokoyama, A. *et al.* The menin tumor suppressor protein is an essential oncogenic cofactor for MLL-associated leukemogenesis. *Cell* **123**, 207–18 (2005).
12. He, S. *et al.* Menin-MLL inhibitors block oncogenic transformation by MLL fusion proteins in a fusion partner independent manner. *Leukemia* **3**, 1–5 (2015).
13. Borkin, D. *et al.* Pharmacologic Inhibition of the Menin-MLL Interaction Blocks Progression of MLL Leukemia In Vivo. *Cancer Cell* **27**, 589–602 (2015).
14. Cermakova, K. *et al.* Validation and Structural Characterisation of the LEDGF/p75-MLL Interface as a New Target for the Treatment of MLL-Dependent Leukaemia. *Cancer Res.* (2014). doi:10.1158/0008-5472.CAN-13-3602
15. Méreau, H. *et al.* Impairing MLL-fusion gene-mediated transformation by dissecting critical interactions with the lens epithelium-derived growth factor (LEDGF/p75). *Leukemia* **27**, 1245–53 (2013).
16. Huang, J. *et al.* The same pocket in menin binds both MLL and JUND but has opposite effects on transcription. *Nature* **482**, 542–6 (2012).
17. Vanderlinden, W., Lipfert, J., Demeulemeester, J., Debyser, Z. & De Feyter, S. Structure, mechanics, and binding mode heterogeneity of LEDGF/p75-DNA nucleoprotein

- complexes revealed by scanning force microscopy. *Nanoscale* 17–19 (2014).
doi:10.1039/c4nr00022f
18. Ruthenburg, A. J., Allis, C. D. & Wysocka, J. Methylation of Lysine 4 on Histone H3: Intricacy of Writing and Reading a Single Epigenetic Mark. *Mol. Cell* **25**, 15–30 (2007).
 19. Southall, S. M., Wong, P.-S., Odho, Z., Roe, S. M. & Wilson, J. R. Structural basis for the requirement of additional factors for MLL1 SET domain activity and recognition of epigenetic marks. *Mol. Cell* **33**, 181–91 (2009).
 20. Mishra, B. P. *et al.* The histone methyltransferase activity of MLL1 is dispensable for hematopoiesis and leukemogenesis. *Cell Rep.* **7**, 1239–47 (2014).
 21. Bernt, K. M. *et al.* MLL-rearranged leukemia is dependent on aberrant H3K79 methylation by DOT1L. *Cancer Cell* **20**, 66–78 (2011).
 22. McLean, C. M., Karemaker, I. D. & van Leeuwen, F. The emerging roles of DOT1L in leukemia and normal development. *Leukemia* **28**, 2131–2138 (2014).
 23. Wong, M., Polly, P. & Liu, T. The histone methyltransferase DOT1L: regulatory functions and a cancer therapy target. *Am. J. Cancer Res.* **5**, 2823–2837 (2015).
 24. Ali, M., Hom, R. a, Blakeslee, W., Ikenouye, L. & Kutateladze, T. G. Diverse functions of PHD fingers of the MLL/KMT2 subfamily. *Biochim. Biophys. Acta* **1843**, 366–71 (2014).
 25. Chang, P.-Y. *et al.* Binding of the MLL PHD3 finger to histone H3K4me3 is required for MLL-dependent gene transcription. *J. Mol. Biol.* **400**, 137–44 (2010).
 26. Mellacheruvu, D. *et al.* The CRAPome: a contaminant repository for affinity purification-mass spectrometry data. *Nat. Methods* **10**, 730–6 (2013).
 27. Lambert, J. P., Tucholska, M., Go, C., Knight, J. D. R. & Gingras, A. C. Proximity biotinylation and affinity purification are complementary approaches for the interactome mapping of chromatin-associated protein complexes. *J. Proteomics* **118**, 81–94 (2015).
 28. Amara, J. F. *et al.* A versatile synthetic dimerizer for the regulation of protein-protein interactions. *Proc. Natl. Acad. Sci. U. S. A.* **94**, 10618–10623 (1997).
 29. Milne, T. A., Martin, M. E., Brock, H. W., Slany, R. K. & Hess, J. L. Leukemogenic MLL fusion proteins bind across a broad region of the Hox a9 locus, promoting transcription and multiple histone modifications. *Cancer Res.* **65**, 11367–11374 (2005).
 30. Dorrance, A. M. *et al.* Mll partial tandem duplication induces aberrant Hox expression in vivo via specific epigenetic alterations. *J. Clin. Invest.* **116**, 2707–2716 (2006).
 31. Andersson, a *et al.* Gene expression profiling of leukemic cell lines reveals conserved molecular signatures among subtypes with specific genetic aberrations. *Leukemia* **19**, 1042–50 (2005).
 32. Kühn, M. W. M. *et al.* MLL partial tandem duplication leukemia cells are sensitive to small molecule DOT1L inhibition. *Haematologica* **100**, e190–e193 (2015).
 33. Ross, M. E. *et al.* Gene expression profiling of pediatric acute myelogenous leukemia. *Gene Expr.* **104**, 3679–3687 (2004).
 34. Lavallée, V.-P. *et al.* The transcriptomic landscape and directed chemical interrogation of MLL-rearranged acute myeloid leukemias. *Nat. Genet.* **47**, (2015).
 35. Yokoyama, A., Lin, M., Naresh, A., Kitabayashi, I. & Cleary, M. L. A higher-order complex containing AF4 and ENL family proteins with P-TEFb facilitates oncogenic and physiologic MLL-dependent transcription. *Cancer Cell* **17**, 198–212 (2010).
 36. Mohan, M. *et al.* Linking H3K79 trimethylation to Wnt signaling through a novel Dot1-containing complex (DotCom). *Genes Dev.* **24**, 574–89 (2010).
 37. Linder, B. *et al.* Biochemical analyses of the AF10 protein: the extended LAP/PHD-finger

- mediates oligomerisation. *J. Mol. Biol.* **299**, 369–78 (2000).
38. Marschalek, R. & Ph, D. Systematic Classification of Mixed-Lineage Leukemia Fusion Partners Predicts Additional Cancer Pathways. *Ann. Lab. Med.* **36**, 85–100 (2016).
 39. Meyer, C. *et al.* The MLL recombinome of acute leukemias in 2013. *Leukemia* **27**, 2165–76 (2013).